# ARS: AUTOMATIC ROUTING SOLVER WITH LARGE LANGUAGE MODELS

## ABSTRACT

Real-world Vehicle Routing Problems (VRPs) are characterized by a variety of practical constraints, making manual solver design both knowledge-intensive and time-consuming. Although there is increasing effort in automating the design of routing solvers, existing research has explored only a limited array of VRP variants and fails to adequately address the complex and prevalent constraints encountered in real-world situations. To fill this gap, we propose the Automatic Routing Solver (ARS), which leverages Large Language Model (LLM) agents to enhance a backbone metaheuristic framework. ARS automatically generates constraint-aware heuristic code from natural language problem descriptions, enabling the framework to handle a wider range of VRP variants without relying on cumbersome modeling rules. Alongside ARS, we introduce RoutBench [1], a benchmark comprising 1,000 VRP variants derived from 24 attributes, designed to rigorously evaluate the effectiveness of automatic routing solvers in handling VRPs with diverse practical constraints. In our experiments, ARS achieves a success rate of over 90% on common VRPs and over 60% on RoutBench, outperforming the other seven LLM-based methods by at least 30% in success rate. Compared to three general-purpose solvers, the ARS framework not only makes it easier for an LLM to generate correct code, with approximately 25% higher correctness, but also achieves superior solving efficiency across many VRP variants.

## 1 INTRODUCTION

The Vehicle Routing Problem (VRP) is a fundamental Combinatorial Optimization Problem (COP) that plays a critical role in logistics, transportation, manufacturing, retail distribution, and delivery planning (Toth & Vigo, 2014; Liu et al., 2023). In these scenarios, the objective of VRPs is to efficiently allocate and plan vehicle routes to meet various requirements while minimizing overall routing costs. These requirements often include constraints such as vehicle capacities, time windows, and duration limits, resulting in numerous variants of VRPs in practical applications (Braekers et al., 2016). However, existing heuristics are usually problem-specific. When the problem changes slightly (e.g., a minor modification to the requirements), a lot of effort is required for experts to redesign the heuristic to make it effective in solving the new problems (Vidal et al., 2013; Rabbouch et al., 2021; Errami et al., 2023).

Large Language Models (LLMs) have shown powerful reasoning and code-generation capabilities (Chen et al., 2021; Austin et al., 2021; Li et al., 2023). By integrating these functionalities, users can express their specific requirements in natural language, enabling the models to automatically design algorithms to address VRPs (Liu et al., 2024c). Most existing works primarily leverage the LLM to solve a small number of standard VRPs (Jiang et al., 2024; Huang et al., 2024) and cannot be applied to complex VRPs. Some recent studies have employed LLMs to formulate routing problems as integer programming models and solve them using general-purpose solvers (e.g., OR-Tools, Gurobi) (Xiao et al., 2023; Zhang et al., 2024; Jiang et al., 2025). However, these approaches may face challenges in adhering to standard modeling practices, and the solvers often exhibit low efficiency, limiting their ability to handle complex real-world constraints.

---

[1] https://anonymous.4open.science/r/RoutBench/

To tackle these challenges, this paper proposes a framework that uses a heuristic algorithm, developed with the assistance of LLMs, to automatically solve the VRP variants with complex constraints. Our contributions are summarized as follows:

- We propose ARS, a framework designed to automatically generate constraint-aware heuristics based on the problem description, which enhances a backbone heuristic algorithm for route optimization, offering an adaptive framework to address the diverse routing problems expressed in natural language.

- We introduce RoutBench, a benchmark with 1,000 VRP variants derived from 24 VRP constraints. Each variant in RoutBench is equipped with a detailed problem description, instance data, and validation code, enabling the evaluation of the effectiveness of various routing solvers in handling diverse VRP constraints.

- We comprehensively validate our approach on common problems and RoutBench. Compared to seven LLM-based methods, ARS achieves a success rate at least 30% higher. Compared to three general-purpose solvers, our framework enables LLMs to generate correct code for various VRPs more easily, without relying on cumbersome modeling rules.

## 2 PROBLEM FORMULATION

Vehicle Routing Problems (VRPs) involve optimizing the routes and schedules of a fleet of vehicles delivering goods or services to various locations, aiming to minimize costs while satisfying constraints like delivery windows and vehicle capacities. The VRP variants can be mathematically described as optimization problems on a graph $\mathcal{G} = (\mathcal{V}, \mathcal{E})$ where nodes $\mathcal{V} = \{0, 1, \ldots, n\}$ represent depot 0 and locations $\{1, \ldots, n\}$, and edges represent the possible routes between these nodes $\mathcal{E} = \{e_{ij}, i, j \in \mathcal{V}\}$, each of them is assigned with a cost $c_{ij}$. The mathematical representation is given by:

$$\min \sum_{i \in V} \sum_{j \in V} c_{ij} x_{ij},$$
$$\text{subject to} \quad \mathbf{x} \in C, \tag{1}$$

where $\mathbf{x} = \{x_{ij} \mid i, j \in \mathcal{V}, \ i \neq j\}$ represents the set of decision variables, $x_{ij}$ is a binary variable that indicates whether the route from $i$ to $j$ is used. The feasible solution space $C$ is defined by constraints. In this paper, we consider VRPs with a variety of real-world constraints such as vehicle capacity, travel distance, and time windows, thereby extending the basic VRP, as seen in the Capacitated VRP (CVRP) (Toth & Vigo, 2014) and the VRP with Time Windows (VRPTW) (Solomon, 1987). Moreover, new constraints often emerge in real-world scenarios. For example, VRP variants related to vehicle capacity include Heterogeneous VRP (HVRP), which considers vehicles with different capacities (Lai et al., 2016), Multi-Product VRP (MPVRP), which addresses the need to transport multiple types of products (Yuceer, 1997), and dynamic demands (Powell, 1986). VRP variants that incorporate these real-world constraints are more prevalent and practically significant in real-world applications.

However, current methods focus on a limited range of problems and do not sufficiently address the complex and diverse constraints present in real-world scenarios. To bridge this gap and further the development of automated solutions for practical VRP variants, this paper introduces a benchmark for VRPs featuring various complex yet practical constraints. Additionally, we propose a general automatic routing solver enhanced by a large language model to effectively manage these constraints.

## 3 AUTOMATIC ROUTING SOLVER

Given the problem description in natural language format and the instance data for any VRP variants with one or more constraints, our proposed Automatic Routing Solver (ARS) can automatically generate the constraint-aware heuristic to enhance a backbone heuristic algorithm. ARS consists of three key components: 1) Pre-defined Database, 2) Constraint-aware heuristic generation, and 3) Augmented heuristic solver.

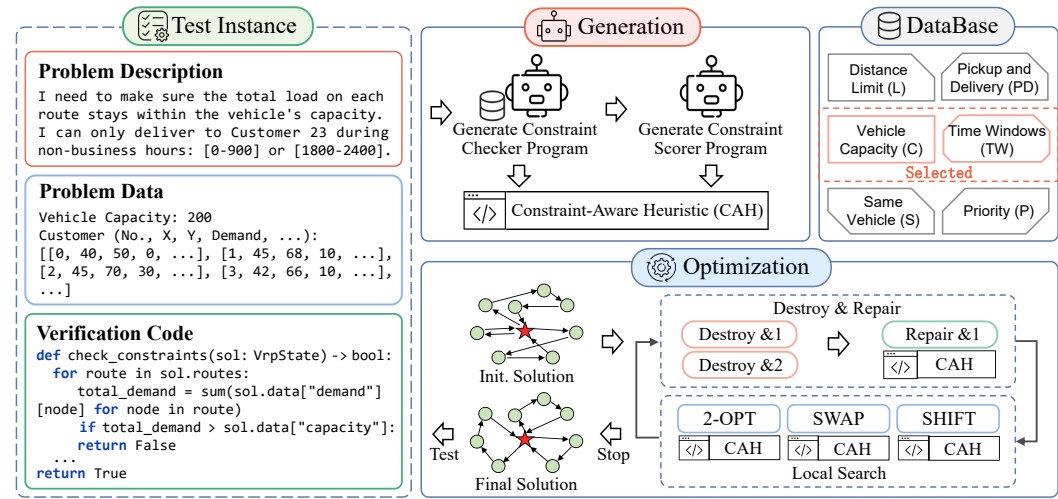

Figure 1: Overview of the proposed ARS framework. The left side of the figure shows the test instance, including the problem description, corresponding data, and validation code for result verification. The right side comprises the database, generation module, and VRP solver. The generation module selects relevant constraints from the database and generates constraint-aware heuristics for the VRP solver to address different VRP variants.

## 3.1 DATABASE

We build a database, denoted as $D$, with several representative fundamental constraints to provide additional guidance for LLM-driven constraint-aware heuristic generation. Specifically, database $D$ includes a basic VRP information $(I_0, C_0)$ (without additional constraints) and six representative constraints $(I_k, C_k)$, $k = 1, \ldots, 6$, each corresponds to a distinct representative constraint: Vehicle Capacity, Distance Limit, Time Windows, Pickup and Delivery, Same Vehicle, and Priority.

Each constraint example $(I_k, C_k)$ consists of two parts:

- $I_k$: The problem description. The natural language description of the constraint.

- $C_k$: The constraint feasibility checking program. It checks whether a solution belongs to the feasible solution space described by $I_k$. The program is given a solution and returns "True" if the corresponding constraint is satisfied.

## 3.2 CONSTRAINT-AWARE HEURISTIC

We first select relative constraints from the database. Then we sequentially generate the constraint checker and scorer programs for the target VRP variants, given the selected constraints and the problem description. Finally, the constraint-aware heuristic is generated based on the designed constraint checker and scorer programs.

### 3.2.1 CONSTRAINT CHECKER PROGRAM GENERATION

Given an input problem description $I$, we instruct an LLM agent to automatically select a subset of relevant constraints $S$ from a database $D$, using them as references to generate the constraint checker program. This process has two steps: 1) Constraint Selection and 2) Constraint Checker Program Generation.

The first step uses a Retrieval-Augmented Generation (RAG) method to retrieve information from the database. There can be two cases. In the first case, LLM agents select one or more relative constraint examples. In the second case, if no constraints are recognized as related to the input $I$, the base case $(I_0, C_0)$, with no additional constraints, is selected.

The second step then works with this set of selected constraints, $S$. For each constraint $C_k$ within $S$, we incorporate specific modifications, $\Delta C_k(I)$, based on the problem description $I$. This process results in the creation of new, customized constraints $C_{new}$, which are better suited to address the specific requirements of the problem. This method offers two main benefits:

- User-Centric Design: It aligns with how users typically work by enhancing existing constraints. This allows users to refine their specific requirements without the need to develop a complete problem definition from the beginning.
- Efficient Processing by LLMs: It helps LLM agents focus on these new, tailored constraints. This reduces unnecessary complexity and enhances the relevance and accuracy of the solutions provided by the models.

### 3.2.2 CONSTRAINT SCORER PROGRAM GENERATION

In practice, heuristic algorithms operate within a variable space that is typically much larger than the feasible solution space. This discrepancy presents significant challenges in identifying feasible solutions. To address this, heuristic methods may permit the presence of infeasible solutions during the search process, as this allows for the evaluation of potential improvements in solution quality (Deb, 2000; Máximo & Nascimento, 2021).

Therefore, to effectively integrate $C_{\text{new}}$ into the solution process, we utilize LLM agents to generate a violation score function guided by the constraint checker program. This score function quantifies the degree of constraint violation, thereby establishing a method for handling constraints and identifying high-quality solutions. It aids in systematically assessing and managing constraint violations, facilitating the search for feasible and high-quality solutions within the expansive variable space.

### 3.2.3 CONSTRAINT-AWARE HEURISTIC GENERATION

We present the Constraint-Aware Heuristic (CAH) based on the constraint checker and scorer programs. As seen in Algorithm 1, the constraint handling method evaluates whether a new solution $s_{new}$ improves upon an old solution $s_{old}$. It first verifies whether $s_{old}$ is feasible (line 1). If $s_{old}$ is feasible, it then verifies whether $s_{new}$ is feasible and has a smaller travel distance (line 2). If both conditions are satisfied, $s_{new}$ is accepted. If $s_{old}$ is infeasible, but $s_{new}$ is either feasible or has a lower violation score than $s_{old}$ (line 5), $s_{new}$ is also accepted. This approach allows infeasible solutions to evolve gradually toward feasibility while minimizing the overall travel distance.

---

**Algorithm 1** Constraint-Aware Heuristic (CAH)

---

**Require:** New solution $s_{new}$, Old solution $s_{old}$
1: **if** Checker($s_{old}$) is feasible **then**
2:     **if** Checker($s_{new}$) is feasible **and** Cost($s_{new}$) < Cost($s_{old}$) **then**
3:         **return true**;
4:     **end if**
5: **else if** Checker($s_{new}$) is feasible **or** Scorer($s_{new}$) < Scorer($s_{old}$) **then**
6:     **return true**;
7: **end if**
8: **return false**;

---

## 3.3 AUGMENTED HEURISTIC SOLVER

The solver has a general single-point-based search backbone heuristic framework, which utilizes automatically generated constraint-aware heuristics to solve various VRP variants. This backbone heuristic solver mainly consists of 1) destroy&repair and 2) local search.

In the destroy phase, we employ multiple operators, including random removal and string removal (Christiaens & Vanden Berghe, 2020), to selectively remove customers or partial routes from the current solution. The choice of destroy operators is determined using a roulette wheel selection mechanism, which assigns higher probabilities to operators that performed well in previous iterations. The repair phase reinserts removed customers into the solution using a greedy repair

operator, aiming to construct a feasible solution with shorter routes. Specific details of these operators are provided in Appendix B.4.

Following the destroy&repair phase, the solution undergoes a local search process to further refine its quality. We utilize a set of local search operators, including 2-OPT (Lin, 1965), SWAP (Osman, 1993), and SHIFT (Rosenkrantz et al., 1977), and the best solution found among these operators is selected. To avoid premature convergence, the Record-to-Record Travel (RRT) criterion is applied (Dueck, 1993; Santini et al., 2018), allowing the acceptance of slightly worse solutions within a predefined threshold, thus maintaining a balance between intensification and diversification. The entire process is iteratively repeated until a termination condition is met, such as a time limit.

Throughout the search, the constraint-aware heuristic serves as the selection strategy for solution updates, applied in both the recreate step and all local search operators. This design allows existing algorithms, which provide the fundamental search capabilities, while the LLM-generated heuristic evaluates and guides this exploration, ensuring the solver not only satisfies user requirements but also effectively minimizes the total route length.

# 4 ROUTBENCH

Recent advancements in LLMs have opened up new possibilities for automatically generating routing solvers to address different VRP variants (Xiao et al., 2023; Chen et al., 2023). However, they are typically evaluated on only a few dozen simple problems, leaving a significant gap in assessing their generalization ability. There is yet to be a VRP benchmark that can evaluate the generalization ability of these methods, especially under complex and diverse constraints in real-world scenarios.

Table 1: A classification of common VRP variants is presented based on six constraint types, with abbreviations provided for both the constraints and VRP variants in parentheses. Each constraint type includes four distinct variants, with examples provided for each, resulting in a total of 24 constraints.

| Basic Constraints | VRP Variants |
| --- | --- |
| Vehicle Capacity (C) | Capacitated VRP (CVRP) (Vidal, 2022; Luo et al., 2023)
Heterogeneous CVRP (HCVRP) (Lai et al., 2016)
Multi-Product VRP (MVRP) (Yuceer, 1997)
Dynamic CVRP (DCVRP) (Powell, 1986) |
| Distance Limit (L) | VRP with Distance Limit (VRPL) (Laporte et al., 1985; Li et al., 1992)
Heterogeneous VRPL (HVRPL) (Lee et al., 2021)
Recharging VRP (RVRP) (Conrad & Figliozzi, 2011; Erdoğan & Miller-Hooks, 2012)
Dynamic VRPL (DVRPL) (Suzuki, 2011; Khouadjia et al., 2012; Qian & Eglese, 2016) |
| Time Windows (TW) | VRP with Time Windows (VRPTW) (Solomon, 1987)
Heterogeneous VRPTW (HVRPTW) (Ren et al., 2010; Vidal et al., 2014)
VRP with Multiple Time Windows (VRPMTW) (Belhaiza et al., 2014)
Dynamic VRPTW (DVRPTW) (Ghiani et al., 2003; Pillac et al., 2013) |
| Pickup and Delivery (PD) | VRP with Mixed Pickup and Delivery (VRPMPD) (Avci & Topaloglu, 2015)
Heterogeneous VRPMPD (HVRPMPD) (Avci & Topaloglu, 2016)
Multi-Product VRPMPD (MVRPMPD) (Zhang et al., 2019)
Dynamic VRPMPD (DVRPMPD) (Gendreau et al., 2006) |
| Same Vehicle (S) | VRP with Same Vehicle Constraint (VRPSVC) (Kumar & Panneerselvam, 2012)
Clustered VRP (CluVRP) (Battarra et al., 2014; Islam et al., 2021)
VRP with Sequential Ordering (VRPSO) (Escudero, 1988; Gambardella & Dorigo, 2000)
VRP with Incompatible Loading Constraint (VRPILC) (Wang et al., 2015) |
| Priority (P) | Precedence constrained VRP (PVRP) (Kubo & Kasugai, 1991)
VRP with Relaxed Priority Rules (VRPRP) (Doan et al., 2021)
VRP with Multiple Priorities (VRPMP) (Yang et al., 2015)
VRP with d-Relaxed Priority Rule (VRP-dRP) (Dasari & Singh, 2023) |

Thus, we propose RoutBench, a benchmark dataset that includes 1,000 VRP variants derived from 24 constraints. These constraints are chosen for their practical significance and theoretical challenges, as highlighted in Table 1. This design not only expands the test scale by two orders of magnitude but also provides an opportunity to evaluate algorithms on unseen VRPs. If an algorithm can effectively solve these unseen problems, it demonstrates the potential to address new real-world challenges, better meeting practical application needs.

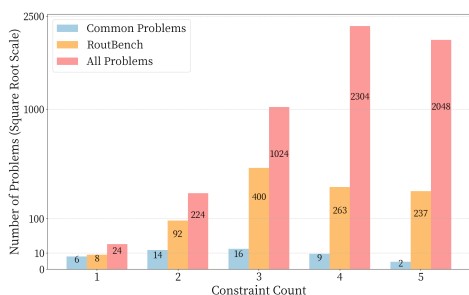
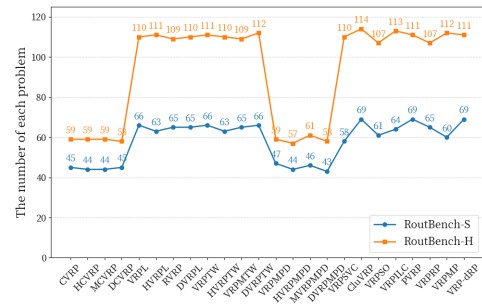

Figure 2: Analysis of problem distribution across common problems, RoutBench, and all problems.

Figure 3: The frequency of constraint usage is analyzed for RoutBench-S and RoutBench-H.

## 4.1 DATASET CONSTRUCTION

The RoutBench is constructed by combinations of six basic constraint types: Vehicle Capacity (C), Distance Limits (L), Time Windows (TW), Pickup and Delivery (PD), Same Vehicle (S), and Priority (P). We design four representative variants derived from each basic constraint type that incorporate variations such as heterogeneous vehicle fleets, multidimensional resource limits, dynamic changes, and others. To produce one problem instance, we first pick one basic constraint combination and then pick one real constraint variant for each chosen basic constraint type. Notice that Vehicle Capacity (C) and Pickup and Delivery (PD) are mutually exclusive, since PD is essentially a derivative of C. The total number of problem combinations is :

$$N_{\text{combinations}} = \sum_{k=1}^{6} \binom{6}{k} \cdot 4^k, \tag{2}$$

where $\binom{6}{k}$ represents the number of ways to select $k$ constraints from the six types, and $4^k$ accounts for the four variations per constraint. After excluding combinations containing both C and PD, the total number of feasible combinations becomes 5624. From these, 1000 unique problem instances are uniformly sampled based on the order of all combinations, ensuring even coverage across the solution space. To balance complexity, RoutBench-S includes 500 problems with three or fewer constraints, while RoutBench-H consists of the remaining 500 problems with more than three constraints.

Each problem instance is comprised of three components: 1) the problem description, which is a natural language explanation of the problem; 2) the instance data, including the geometric positions of nodes and the constraint parameters, with data generated using the Solomon C103 dataset (Solomon, 1987) as a base; and 3) the validation code, used to confirm whether a solution adheres to user requirements and satisfies all constraints. Problem sizes include 25, 50, and 100 nodes. All instances in RoutBench come with Best Known Solutions (BKS), with details provided in Appendix E.6.

The detailed descriptions of the 24 problem instances are provided in Table 6. These examples illustrate the diversity of problem settings and serve as a representative subset of the dataset's broader scope. By systematically combining constraints, leveraging validation mechanisms, and ensuring feasibility, RoutBench offers a diverse and reliable dataset for benchmarking VRP solvers.

## 4.2 ANALYSIS

This section analyzes the distribution of problem types and complexities in the RoutBench dataset, focusing on the 48-problem subset and RoutBench, and their relationship to the full set of 5624 feasible problems.

The distribution of problems by the number of constraints is shown in Figure 2. The 48-problem subset consists of common problems, one is a simple VRP without any constraints, while the remaining 47 include one to five basic constraints. In RoutBench, the distribution reflects the proportions of the full set of 5624 problems. Specifically, problems with three constraints are the most common in the dataset, as three-constraint problems dominate the total number of problems with three or

fewer constraints. For more complex problems, those with four and five constraints appear in similar proportions, ensuring a balanced representation of high-complexity scenarios.

Figure 3 shows the distribution of problem types across RoutBench. Most problem types, such as VRPTW, HVRPL, and CluVRP, are well-represented. However, problems involving vehicle capacity constraints (e.g., CVRP, HCVRP) and pickup and delivery operations (e.g., VRPMPD, HVRPMPD) are less frequent. This is because these two categories are mutually exclusive, and the two types do not coexist in the dataset.

Overall, the RoutBench dataset achieves a diverse and balanced representation of problem types and complexities. The 48-problem subset provides a concise overview of simpler cases, while RoutBench captures a wide range of scenarios.

## 5 EXPERIMENTS

To evaluate the performance of LLM-based methods in handling different VRP variants, we assess their ability to generate correct programs within our solver. To show the difference between our ARS framework and general-purpose solvers, we compare our solver with others (e.g., CPLEX, OR-Tools, and Gurobi) on the success rate of generating programs and the performance of solving various VRP variants. Additionally, we experiment with other closed- and open-source LLMs (e.g., DeepSeek V3 and LLaMA 3.1 70B) to see their impact on program generation. Finally, we conduct an ablation study on our proposed solver.

### 5.1 COMPARISON WITH LLM-BASED METHODS

To evaluate the ability of ARS to generate successful programs, we compare it with seven other LLM-based methods: Standard Prompting, Chain of Thought (CoT) (Wei et al., 2022), Reflexion (Shinn et al., 2024), Progressive-Hint Prompting (PHP) (Zheng et al., 2023), Chain-of-Experts (CoE) (Xiao et al., 2023), Self-debug (Chen et al., 2023), and Self-verification (Huang et al., 2024). To focus on program generation, all methods use our backbone framework to handle the various VRP variants.

We conduct this experiment on 48 common problems and RoutBench using GPT-4o. We use two metrics to evaluate the results. The Success Rate (SR) measures the proportion of programs where the generated solutions pass the validation process. The Runtime Error Rate (RER) is the percentage of programs that fail due to runtime errors, incorrect API usage, or syntax mistakes.

Table 2: The performance comparison between ARS and seven LLM-based methods on common problems and RoutBench. The best results among these methods are highlighted in grey.

| Methods | Common Problems | | RoutBench | | | |
| | | | RoutBench-S | | RoutBench-H | |
| | SR ↑ | RER ↓ | SR ↑ | RER ↓ | SR ↑ | RER ↓ |
|---|---|---|---|---|---|---|
| Standard Prompting | 41.67% | 6.25% | 37.60% | 8.20% | 11.60% | 15.80% |
| CoT | 37.50% | 8.33% | 37.40% | 9.60% | 13.40% | 14.80% |
| Reflexion | 45.83% | 6.25% | 41.80% | 4.60% | 15.20% | 7.60% |
| PHP | 33.33% | 10.42% | 32.60% | 12.00% | 11.20% | 17.40% |
| CoE | 37.50% | 2.08% | 34.20% | 8.00% | 11.40% | 12.20% |
| Self-debug | 43.75% | 0.00% | 34.60% | 4.00% | 10.80% | 7.60% |
| Self-verification | 43.75% | 2.08% | 34.00% | 5.20% | 15.60% | 9.20% |
| ARS (Ours) | 91.67% | 0.00% | 73.20% | 5.20% | 46.80% | 11.80% |

As shown in Table 2, ARS significantly outperforms other LLM-based methods in generating correct programs for both common problems and RoutBench. It is evident that these compared methods exhibit unsatisfactory performance across all problems, particularly on the RoutBench-H problems, where their SR is merely around 10%. None of the compared algorithms achieved an SR of 50% or higher. In contrast, ARS achieved an SR of 91.67% on the common problems. Moreover, on RoutBench, ARS generated correct solutions for 60% of the VRP variants, outperforming all seven other LLM-based methods by at least 30% in terms of SR. These results highlight the generality of ARS in addressing complex VRP variants.

## 5.2 Comparison with Different Solvers

To evaluate the difference of LLM in code generation between our solver and general-purpose solvers, we ask an LLM using only Standard Prompting to generate constraint-handling code for four solvers (i.e., CPLEX, OR-Tools, Gurobi, and our solver). As shown in Figures 4 and 5, under standard prompting, our solver achieves the highest Success Rate on the common problems and requires the fewest Lines of Code compared to other solvers. This is due to two advantages of our framework. First, the LLM only needs to convert the problem description into constraint-handling code using straightforward Python syntax, while other solvers require highly standardized modeling. Second, our framework allows the LLM to focus solely on generating the constraint-handling code, which is then used directly by our solver, rather than requiring the LLM to understand the entire framework first. For the same VRP variants, our framework makes program generation easier compared to these solvers. Examples of solver codes are provided in Appendix F.

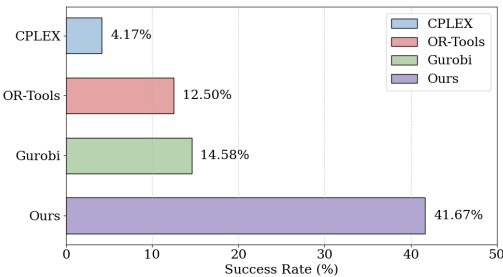

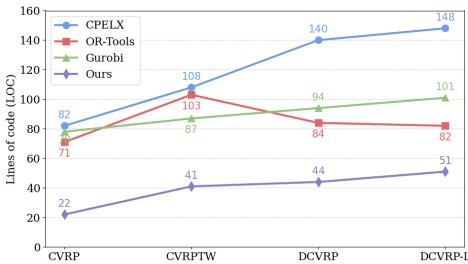

Figure 4: Comparison of the success rate for different solvers with standard prompting.

Figure 5: Compares the lines of code generated by LLMs for different solvers on four VRP variants.

To explore the performance of these solvers on different VRP variants, we test them on four VRP variants, including two classic variants, CVRP and CVRPTW, as well as two dynamic variants, DCVRP and DCVRP-L. Details of these problems are provided in Appendix E.3. Since general-purpose solvers often require considerable time for VRPs, we allocated time limits of 25, 50, and 100 minutes for problem sizes of 25, 50, and 100, respectively. As shown in Table 1, our solver achieved the optimal results in all tested instances. We also tested it on instances with up to 200 nodes from the CVRPLIB benchmark, where our solver still performed best among the four solvers (results are in Appendix E.10). Notably, our framework is designed to enable a search algorithm to handle a wide range of VRPs, rather than to claim state-of-the-art performance on a few specific problems. This is why the results for other methods (e.g., HGS and LKH-3) are provided for reference only.

Table 3: Performance analysis of different solvers. The table presents the gaps compared to the results obtained by ARS. The record time is the time to find the best solution per run. The best results among the four solvers (i.e., CPLEX, OR-Tools, Gurobi, and ARS) are highlighted in grey.

| Solvers | CVRP | | | | | | | | | CVRPTW | | | | | | | | |
|---|---|---|---|---|---|---|---|---|---|---|---|---|---|---|---|---|---|---|
| | 25 Nodes | | | 50 Nodes | | | 100 Nodes | | | 25 Nodes | | | 50 Nodes | | | 100 Nodes | | |
| | Obj. ↓ | Gap | Time | Obj. ↓ | Gap | Time | Obj. ↓ | Gap | Time | Obj. ↓ | Gap | Time | Obj. ↓ | Gap | Time | Obj. ↓ | Gap | Time |
| HGS | 186.9 | 0.00% | 0.1s | 358.0 | 0.00% | 0.2s | 817.8 | 0.00% | 0.3s | 190.3 | 0.00% | 0.1s | 361.4 | 0.00% | 0.1s | 826.3 | 0.00% | 0.4s |
| LKH-3 | 186.9 | 0.00% | 0.1s | 358.0 | 0.00% | 0.1s | 817.8 | 0.00% | 1.3s | 190.3 | 0.00% | 0.1s | 361.4 | 0.00% | 0.1s | 826.3 | 0.00% | 4.2s |
| CPLEX | 187.6 | 0.37% | 45s | 362.7 | 1.31% | 47m | 841.6 | 2.91% | 1.1h | 190.3 | 0.00% | 1.3s | 361.4 | 0.00% | 5.2m | 826.3 | 0.00% | 1.3h |
| OR-Tools | 186.9 | 0.00% | 4.2s | 358.8 | 0.22% | 7.2s | 849.2 | 3.84% | 1.0m | 190.3 | 0.00% | 0.5s | 362.5 | 0.30% | 9.6s | 828.1 | 0.21% | 3.1m |
| Gurobi | 186.9 | 0.00% | 1.8m | 358.0 | 0.00% | 5.6m | 828.0 | 1.25% | 1.3h | 190.3 | 0.00% | 10s | 361.4 | 0.00% | 1.2m | 826.3 | 0.00% | 1.4h |
| Ours | 186.9 | 0.00% | 2.3s | 358.0 | 0.00% | 19.1s | 817.8 | 0.00% | 2.4m | 190.3 | 0.00% | 3.7s | 361.4 | 0.00% | 10.2s | 826.3 | 0.00% | 4.3m |

| Solvers | DCVRP | | | | | | | | | DCVRP-L | | | | | | | | |
|---|---|---|---|---|---|---|---|---|---|---|---|---|---|---|---|---|---|---|
| | 25 Nodes | | | 50 Nodes | | | 100 Nodes | | | 25 Nodes | | | 50 Nodes | | | 100 Nodes | | |
| | Obj. ↓ | Gap | Time | Obj. ↓ | Gap | Time | Obj. ↓ | Gap | Time | Obj. ↓ | Gap | Time | Obj. ↓ | Gap | Time | Obj. ↓ | Gap | Time |
| CPLEX | 213.3 | 9.16% | 35s | 392.1 | 6.98% | 50m | 878.4 | 5.65% | 1.2h | 215.3 | 0.00% | 1.6m | 390.7 | 1.11% | 48m | 871.0 | 2.68% | 1.6h |
| OR-Tools | 253.4 | 29.68% | 0.7s | 426.0 | 16.23% | 39s | 893.6 | 7.48% | 1.4h | 226.3 | 5.11% | 1.4s | 405.8 | 5.02% | 1.5m | 874.6 | 3.10% | 11m |
| Gurobi | 219.2 | 12.18% | 5.2m | 395.1 | 7.80% | 40m | 874.8 | 5.22% | 1.1h | 219.2 | 1.81% | 2.4m | 395.5 | 2.36% | 44m | 876.8 | 3.36% | 1.0h |
| Ours | 195.4 | 0.00% | 6s | 366.5 | 0.00% | 1.8m | 831.4 | 0.00% | 18m | 215.3 | 0.00% | 2.4s | 386.4 | 0.00% | 26s | 848.3 | 0.00% | 19m |

## 5.3 EVALUATION WITH DIFFERENT LLMS

We further investigate the impact of using different LLMs (i.e., GPT-3.5-Turbo, GPT-4o, DeepSeek-V3, and LLaMA-3.1-70B) for generating correct programs to solve VRP variants under standard prompting and ARS. As shown in Figure 6, all methods benefit from more advanced LLMs, leading to improved accuracy. The results indicate that DeepSeek-V3 is the most effective LLM for handling VRP variants among these LLMs, and ARS achieves an SR of 77.20% on simple problems in RoutBench. The improvement is even more pronounced for RoutBench-H problems, where ARS attains an SR of 61.60%. Further discussion can be found in Appendix E.4.

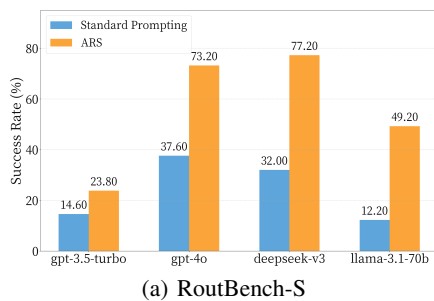

(a) RoutBench-S

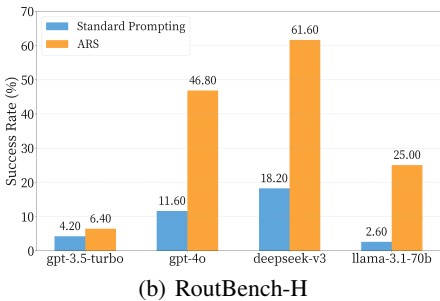

(b) RoutBench-H

Figure 6: The performance of standard prompting and ARS is compared across various LLMs. ARS demonstrates compatibility with different models and shows clear advantages in RoutBench.

## 5.4 ABLATION STUDY

We conducted an ablation study on ARS by removing its database and constraint selection components individually. As shown in Table 4, removing constraint selection decreases the success rate (SR). This is because, without this step, ARS uses all six representative constraints, which can mislead the LLM with irrelevant information.

The impact is more significant when removing the database. This effectively reverts the process to Standard Prompting, forcing the LLM to generate all constraints independently and thus increasing the difficulty. These results demonstrate that both components are essential for ARS to achieve optimal performance. A more detailed discussion is provided in Appendix E.7.

Table 4: Ablation study on ARS for program generation. The best results among these methods are highlighted in grey.

| Methods | Common Problems | |
| --- | --- | --- |
| | SR ↑ | RER ↓ |
| w/o Constraint Selection | 62.50% | 2.08% |
| w/o Database | 41.67% | 6.25% |
| ARS (full) | 91.67% | 0.00% |

## 6 CONCLUSION, LIMITATION, AND FUTURE WORK

**Conclusion.** In this paper, we propose ARS, a framework that leverages LLMs to automatically generate constraint-aware heuristics that enhance a backbone algorithm for a wide range of VRPs. We also introduce RoutBench, a comprehensive benchmark of 1,000 VRP variants derived from 24 distinct constraints. Each variant provides a problem description, instance data, and validation code to facilitate the standardized evaluation of routing solvers. Our results show that compared to general-purpose solvers, ARS not only enables an LLM to generate correct code more easily but also achieves superior solving efficiency across many VRP variants.

**Limitation and Future Work.** In this paper, we focus on leveraging LLMs to enable an existing search algorithm to handle a wide range of VRPs. In the future, this work can be extended by either refining the underlying search algorithm to enhance the search capabilities of the solver, or by replacing it to apply the framework to other domains, such as 3D bin packing.

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

This is an appendix for "ARS: Automatic Routing Solver with Large Language Models". Specifically, we provide:

- Related works on heuristics, NCO, and LLMs for VRPs (Appendix A).

- Detailed explanation of the methodology, including the prompts used in ARS, examples of LLM outputs, and the operators employed (Appendix B).

- Function template for the constraint Checker and scorer programs (Appendix C).

- Details of 48 common problems and constraints for 24 VRP variants (Appendix D).

- More experimental results and analyses, including ARS analysis, LLM-suggested constraints, enhancements, and CVRPLib evaluation (Appendix E).

- Code examples for our solver, CPLEX, OR-Tools, and Gurobi (Appendix F).

- The potential societal impact of this work (Appendix G).

- A statement on the use of Large Language Models for manuscript preparation (Appendix H).

- The licenses and URLs of the baseline methods (Appendix I).

## A    RELATED WORKS

The Vehicle Routing Problem (VRP) is a classical combinatorial optimization problem that seeks optimal routes for vehicles to serve customers under constraints like capacity and time windows (Dantzig & Ramser, 1959). Over the years, many VRP variants have been developed and extensively studied, including the Capacitated VRP (CVRP) (Lysgaard et al., 2004), the VRP with Time Windows (VRPTW) (Solomon, 1987), and the Multi-Depot VRP (MDVRP) (Yuceer, 1997). However, solving these VRP variants often requires experts to deeply understand the specific problem characteristics, including the constraints, customer demands, and operational rules. This process involves carefully analyzing the problem, designing appropriate models, and implementing customized algorithms. Such methods usually require complex coding and are limited in their ability to address only a small number of VRP variants, making them less flexible and scalable for the diverse and evolving challenges in real-world applications.

### A.1    HEURISTICS FOR VRPS

Traditional methods for solving VRP often rely on heuristics. Some simple heuristics, such as the Greedy algorithm and hill-climbing, are commonly used to solve VRP. The Greedy algorithm builds a solution step-by-step by making the most immediate, optimal choice at each step, though it often leads to suboptimal global solutions (Shafique & Shah, 2005). Hill-climbing, on the other hand, iteratively improves a solution by moving to a better neighboring solution, but it is prone to getting stuck in local optima (Gent & Walsh, 1993). To address these limitations, more advanced metaheuristics have been developed. Simulated Annealing (SA) probabilistically accepts worse solutions during the search process to escape local optima, mimicking the physical annealing process (Kirkpatrick et al., 1983). Tabu Search (TS) enhances local search by using a tabu list to prevent revisiting previously explored solutions, enabling it to explore broader solution spaces (Glover, 1989).

In addition to these, state-of-the-art approaches like the Hybrid Genetic Search (HGS) and Lin-Kernighan-Helsgaun (LKH-3) algorithms have achieved remarkable success in solving VRPs. HGS combines genetic algorithms with heuristics tailored for specific types of VRPs, efficiently balancing exploration and exploitation (Vidal et al., 2013). It is particularly powerful for large-scale and complex VRPs. LKH-3, an extension of the classic Lin-Kernighan heuristic, is highly effective for solving Traveling Salesman Problems (TSP) and TSP-based VRPs, leveraging advanced search strategies and efficient implementations to achieve near-optimal solutions (Helsgaun, 2017).

Adaptive Large Neighborhood Search (ALNS) represents a more dynamic and flexible approach (Ropke & Pisinger, 2006). It adaptively selects different neighborhood operators based on their performance during the search process, making it highly effective for solving complex VRP variants (Mara et al., 2022). Recent advancements in ALNS (Wang et al., 2024) integrate machine learning techniques to predict the most effective operators and reinforcement learning to optimize

selection policies. Hybrid ALNS approaches, such as combining ALNS with branch-and-price methods, further enhance their ability to solve constrained and large-scale VRPs (Vidal, 2022).

Despite their success, these methods often require experts to deeply understand the specific VRP variant, carefully model the problem, and implement customized algorithms. This reliance on expert knowledge limits the scalability of these approaches to handle a broader range of VRP variants without significant manual effort.

## A.2 NCO FOR VRPS

Neural combinatorial optimization (NCO) represents a paradigm shift in solving VRPs by leveraging deep learning models to directly learn problem-solving strategies from data. Several notable NCO approaches have been proposed to address the challenge of solving multiple VRP variants within a unified framework. For instance, MTPOMO tackles cross-problem generalization by representing VRPs as combinations of shared attributes, allowing a single model to solve unseen variants in a zero-shot manner (Liu et al., 2024a). MVMoE employs a multi-task learning framework with a mixture-of-experts architecture, using a hierarchical gating mechanism to balance model capacity and computational efficiency, achieving strong results across ten unseen VRP variants (Zhou et al., 2024). CaDA further advances the field by incorporating a constraint-aware dual-attention mechanism, which effectively captures both global and local problem-specific information, enabling state-of-the-art performance on sixteen VRP variants (Li et al., 2024).

However, despite these advancements, current NCO methods still face significant limitations. They typically require manual modifications to adapt algorithms for new VRP variants, limiting their scalability and practicality for real-world applications with highly diverse constraints.

## A.3 LLMS FOR VRPS

Recent advancements in large language models (LLMs) have introduced new possibilities for solving vehicle routing problems (VRPs) by leveraging their capacity to encode and process complex optimization tasks (Huang et al., 2024). LLM-based automatic heuristic design (AHD) has emerged as a promising approach, enabling the generation of high-quality heuristics for problems like the traveling salesman problem (TSP) and capacitated VRP (CVRP) without extensive domain expertise. Methods such as Evolutionary Optimization Heuristics (EoH) integrate LLMs with evolutionary computation (EC) to iteratively refine a population of heuristics, facilitating automated discovery of effective solutions (Liu et al., 2024b). However, population-based approaches often converge prematurely to local optima. To overcome this, Monte Carlo Tree Search-based AHD (MCTS-AHD) organizes LLM-generated heuristics into a tree structure, enabling deeper exploration of the search space and better utilization of underperforming heuristics (Zheng et al., 2025).

Other studies have explored the application of LLMs to different VRP variants, showcasing innovative approaches and promising results. For example, LLM-driven Evolutionary Algorithms (LMEA) utilize LLMs as evolutionary optimizers, achieving competitive results on TSPs with minimal domain knowledge (Liu et al., 2024d). Mechanisms like self-adaptation help balance exploration and exploitation, effectively avoiding local optima. Similarly, an approach proposed by Huang et al. (Huang et al., 2024) enables LLMs to directly generate executable programs for VRPs from natural language task descriptions. This method is further enhanced by a self-reflection framework, which allows LLMs to debug and verify their solutions, significantly improving feasibility, optimality, and efficiency. These early explorations highlight the potential of LLMs in addressing VRPs and advancing the field.

Another line of research explores transforming textual problem descriptions into mathematical formulations and executable code that can be processed by external solvers (Tang et al., 2024). This approach benefits from LLMs' ability to interpret user queries and generate structured outputs, enabling the automation of optimization tasks. In parallel, multi-agent systems have been introduced to coordinate LLM-based agents for tasks such as problem formulation, programming, and evaluation (Xiao et al., 2023). Separately, the DRoC framework introduces a novel method for solving complex VRPs by decomposing constraints, retrieving external knowledge through a retrieval-augmented generation (RAG) approach, and integrating it with the model's internal knowledge. By dynamically optimizing program generation, this framework has demonstrated significant improvements in both accuracy and

runtime efficiency across 48 VRP variants (Jiang et al., 2025). However, despite these innovations, these methods remain inherently constrained by the scope of knowledge encoded within pre-trained models, particularly in generating solver-specific code. This limitation poses significant challenges for LLMs in addressing novel or highly complex problems (Zhang et al., 2024).

# B  DETAILED METHODOLOGY

## B.1  PROMPTS OF ARS

Automatic Routing Solver (ARS) is designed to address each VRP variant by leveraging LLMs in two key steps: **Constraint Checking Program Generation** and **Constraint Scoring Program Generation**, with a total of three LLM calls. In this section, we describe the prompt engineering involved in each step. These prompts are constructed based on user inputs and several representative constraints stored in the database to generate the *Constraint Checking Program* and the *Constraint Scoring Program*. Variable information, such as user inputs and constraint descriptions, is highlighted in blue for clarity.

**Step 1.1: Constraint Selection.** In the first step, the ARS identifies constraints relevant to the user's input from the database. This step processes the user input and matches it against the constraints stored in the database. If relevant constraints are found, they are selected for further processing. Otherwise, LLM outputs *"No Relevant Constraint"*. This step ensures that only the constraints relevant to the problem description are considered in subsequent steps.

---

**Prompt for Constraint Selection**

For the description in the VRP problem, identify and provide the relevant constraint types from the following list:
{constraint_description}

According to the user input:
{user_input}
If no constraint types match the user input, respond with: "No Relevant Constraint".

Do not give additional explanations.

---

**Step 1.2: Constraint Checking Program Generation.** Based on the relevant constraints selected in the previous step, the ARS uses the LLM to generate a new Constraint Checking Program by taking the selected constraints as references. Specifically, the selected constraints are provided as input to the LLM, which then generates the new program tailored to the problem description and the referenced constraint information.

---

**Prompt for Constraint Checker Program Generation**

As a Python algorithm expert, please implement a function to check the constraints for the vehicle routing problem (VRP) based on the provided description and relevant code.

User input:
{user_input}

Relevant Examples:
{related_constraints_and_codes}

Do not give additional explanations.

---

**Step 2: Constraint Scoring Program Generation.** In the final step, the ARS generates a *Constraint Scoring Program* using the *Constraint Checking Program* developed in the previous step as a

foundation. This scoring program evaluates the degree to which the constraints are satisfied by assigning a quantitative score based on the results of the constraint checks.

---

**Prompt for Constraint Scorer Program Generation**

As a Python algorithm expert, please implement a function to calculate the constraint violation score for the Vehicle Routing Problem (VRP) based on the given constraints.

Function Template:
{function_template}

Constraints Description:
{constraints_description}

Check Constraints Code:
{related_constraints_and_codes}

Do not give additional explanations.

---

## B.2 EXAMPLES OF LLM OUTPUTS

This subsection shows an example of using LLM with ARS to solve the CVRP with Incompatible Loading Constraints (CVRP-ILS). The process includes three LLM calls. The first call selects constraints related to CVRP-ILS. The second call creates a program to check these constraints. The third call creates a program to calculate violation scores.

---

### Example of Step 1.1: Constraint Selection

**First Call Input:**
For the description in the VRP problem, identify and provide the relevant constraint types from the following list:
1. No Relevant Constraint
Example: No relevant constraint.
2. Vehicle Capacity Constraint
Example: The total load of each route must not exceed the vehicle capacity.
3. Distance Limit Constraint
Example: Each route's total length must not exceed 100 units.
4. Time Windows Constraint
Example: The arrival time at the node must meet the time windows.
5. Pickup and Delivery Constraint
Example: At Node [24], 10 units of goods are picked up. Unlike delivery, visiting a pickup node reduces the vehicle's available capacity.
6. Same Vehicle Constraint
Example: Nodes [1, 10] must be served by the same vehicle.
7. Priority Constraint
Example: Nodes [17, 19] are priority points.

According to the user input:
The total load of each route must not exceed the vehicle capacity. Nodes [7, 8] must not be on the same route.
If no constraint types match the user input, respond with: "No Relevant Constraint".

Do not give additional explanations.

**First Call Output:**
————————————— Relevant Constraints Type —————————————
1. Constraint type: Vehicle Capacity Constraint
2. Constraint type: Same Vehicle Constraint

---

The problem description for CVRP-ILS is: The total load of each route must not exceed the vehicle capacity. Nodes [7, 8] must not be on the same route.

In the first step, the relevant constraints for CVRP-ILS are identified and selected. Therefore, the set of constraints for CVRP-ILS is $S = \{$"Vehicle Capacity Constraint", "Same Vehicle Constraint"$\}$. These constraints will guide the next step in helping ARS generate constraint checking program for CVRP-ILS.

**Example of Step 1.2: Constraint Checker Program Generation**

**Second Call Input:**
As a Python algorithm expert, please implement a function to check the constraints for the vehicle routing problem (VRP) based on the provided description and relevant code.

User input:
The total load of each route must not exceed the vehicle capacity. Nodes [7, 8] must not be on the same route.

Relevant Examples:
# See Appendix B.3.

Do not give additional explanations.

**Second Call Output:**

```python
def check_constraints(solution: VrpState) -> bool:
  """
  Check if the solution satisfies the constraints.

  Args:
    solution (VrpState): An object representing the VRP solution.
      - problem_data (dict): A dictionary with problem details:
        - "edge_weight": A 2D NumPy array of node distances .
        - "demand": A 1D NumPy array of node demands (0 for the
            depot).
        - "capacity": Maximum load capacity per vehicle.

  Return:
    bool: True if the solution satisfies the constraints, False
        otherwise.

  Note:
    The above "Args" cannot be added or modified, and no data
        outside the "solution" object should be used or added.
  """
  # Check vehicle capacity constraint
  for route in solution.routes:
    total_demand = sum(solution.problem_data["demand"][node] for
        node in route)
    if total_demand > solution.problem_data["capacity"]:
      return False

  # Check nodes [7, 8] not on same route constraint
  for route in solution.routes:
    if 7 in route and 8 in route:
      return False

  return True
```

**Example of Step 2: Constraint Scorer Program Generation**

**Third Call Input:**

As a Python algorithm expert, please implement a function to calculate the constraint violation score for the Vehicle Routing Problem (VRP) based on the given constraints.

Function Template:
# See Appendix C.

Constraints Description:
The total load of each route must not exceed the vehicle capacity. Nodes [7, 8] must not be on the same route.

Check Constraints Code:
# Use the constraint checking code generated in Step 1.2.

Do not give additional explanations.

**Third Call Output:**

```python
def calculate_violation_score(solution: VrpState) -> float:
    """
    Args:
      solution (VrpState): An object representing the VRP solution.
        - problem_data (dict): A dictionary with problem details; Use
            only defined values in "Check Constraints Code".
        - routes (list): A list of routes, where each route is a list
            of node IDs (integers, excluding the depot).
        - cost (Callable[[], float]): Implemented method to compute
            and return the total cost of the solution's routes.
    Return:
      violation_score (float): The calculated constraint violation
          score, where higher scores indicate more severe violations.

    Note:
      Do not use data or methods that is not contained in the "
          solution" object.
    """
    violation_score = 0.0

    # Check vehicle capacity constraint
    for route in solution.routes:
      total_demand = sum(solution.problem_data["demand"][node] for
          node in route)
      if total_demand > solution.problem_data["capacity"]:
        violation_score += (total_demand - solution.problem_data["
            capacity"])

    # Check nodes [7, 8] not on same route constraint
    for route in solution.routes:
      if 7 in route and 8 in route:
        violation_score += 1.0

    return violation_score
```

## B.3 EXAMPLES OF RELEVANT CONSTRAINT

**Vehicle Capacity:** The total load of each route must not exceed the vehicle capacity.

**Verification Code:**

```python
def check_constraints(solution: VrpState) -> bool:
    """
    Check if the solution satisfies the constraints.

    Args:
      solution (VrpState): An object representing the VRP solution.
        - problem_data (dict): A dictionary with problem details:
          - "edge_weight": A 2D NumPy array of node distances.
          - "demand": A 1D NumPy array of node demands (0 for the depot).
          - "capacity": Maximum load capacity per vehicle.
        - routes (list): A list of routes, where each route is a list of
      node IDs (integers, excluding the depot node 0).

    Return:
      bool: True if the solution satisfies the constraints, False
      otherwise.

    Note:
      The above "Args" cannot be added or modified, and no data outside
      the "solution" object should be used or added.
    """
    for route in solution.routes:
        total_demand = sum(solution.problem_data["demand"][node] for node in
         route)
        if total_demand > solution.problem_data["capacity"]:
            return False
    return True
```

**Same Vehicle:** Nodes [1, 10] must be served by the same vehicle.

**Verification Code:**

```python
def check_constraints(solution: VrpState) -> bool:
    """
    Check if the solution satisfies the constraints.

    Args:
      solution (VrpState): An object representing the VRP solution.
        - problem_data (dict): A dictionary with problem details:
          - "edge_weight": A 2D NumPy array of node distances.
        - routes (list): A list of routes, where each route is a list of
      node IDs (integers, excluding the depot node 0).

    Return:
      bool: True if the solution satisfies the constraints, False
      otherwise.

    Note:
      The above "Args" cannot be added or modified, and no data outside
      the "solution" object should be used or added.
    """
    for route in solution.routes:
      if 1 in route and 10 in route:
        break
    else:
      # If no route contains both nodes 1 and 10
      return False

    return True
```

## B.4 OPERATOR

Local search operators are essential components of heuristic and metaheuristic methods, designed to explore the neighborhood of a solution and iteratively improve its quality. These operators are the building blocks for efficiently navigating the search space, balancing exploration and exploitation. The commonly used operators are as follows:

**2-opt Operator (Lin, 1965).** The 2-opt operator is a classical approach originally developed for the Traveling Salesman Problem (TSP). It works by removing two non-adjacent edges in the solution and reconnecting them in a different way, thereby altering the order of nodes. If the new configuration reduces the total cost, it is accepted as an improved solution.

**Swap Operator (Osman, 1993).** The Swap operator is another simple yet powerful tool in local search methods. It works by exchanging the positions of two elements within the solution. This operation generates a new configuration, which can help in escaping local optima and promoting diversity in the solution space.

**Shift Operator (Rosenkrantz et al., 1977).** The Shift operator involves moving an element from one position in the solution to another. This operation changes the relative ordering of elements, redistributing their positions to explore alternative configurations. By shifting elements, the algorithm can adjust the structure of the solution in a more targeted manner, allowing it to overcome local optimality and discover new regions of the solution space.

**Destroy Operator.** The Destroy operator partially disrupts the current solution by selectively removing some elements. This disruption breaks the local optimality of the solution, allowing for the exploration of new regions in the search space. There are two common implementations of this operator: **Random Removal** and **String Removal**.

- **Random Removal**: This method involves removing elements uniformly at random, without relying on specific heuristics or problem-dependent strategies, making it a straightforward yet highly effective approach to diversify the search process and introduce variability into the solution space.
- **String Removal**: This method targets sequences of consecutive or related elements (strings), such as partial routes or groups of customers Christiaens & Vanden Berghe (2020). It begins by randomly selecting a "center" customer and removing a string of nearby customers from the route. The string size is randomly determined, constrained by the average route size and a predefined maximum. If constraints on the number of disrupted routes or previously disrupted routes are met, further removal is skipped.

**Repair Operator.** The Repair operator complements the Destroy operator by reinserting removed elements to reconstruct a complete solution, guided by optimization objectives. This combination of destruction and repair allows the algorithm to iteratively refine solutions while maintaining the flexibility to explore new possibilities. One commonly used implementation of the Repair operator is **Greedy Repair**:

- **Greedy Repair**: This method reinserts removed elements one by one, selecting at each step the position that minimizes the objective function. By considering constraint-aware heuristics during the reinsertion process, it ensures that each step improves solution quality while adhering to problem-specific constraints, effectively balancing optimality and feasibility throughout the search process.

In summary, the local search operators discussed above, including 2-opt, Swap, Shift, Destroy, and Repair, play a crucial role in the design of heuristic and metaheuristic algorithms. These operators enable targeted adjustments to the solution, facilitating efficient exploration and exploitation of the solution space. By combining these operators, algorithms can effectively escape local optima and converge toward high-quality solutions.

In our approach, we adopt these efficient operators within a Backbone heuristic framework, which provides the structural foundation for solving complex optimization problems. The framework leverages these operators to iteratively refine solutions, balancing between intensification and diversification.

## C  FUNCTION TEMPLATE

The following code template serves as a framework to define the checker and scorer functions, ensuring clarity and proper parameter usage. Within the template, the function name is predefined, and the roles of relevant parameters are described in detail. The specific implementation details are generated by the LLM based on the target VRP variant.

**Function template for the Constraint Checker:**

```python
def check_constraints(solution: VrpState) -> bool:
    """
    Check if the solution satisfies the constraints.

    Args:
      solution (VrpState): An object representing the VRP solution.
        - problem_data (dict): A dictionary with problem details:
          - "demand": A 1D NumPy array of node demands (0 for the depot).
          - "capacity": Maximum load capacity per vehicle.
          - "edge_weight": A 2D NumPy array of node distances.
          - "service_time": A 1D NumPy array of node service times (0 for
      the depot).
          - "time_window": A list of [earliest start, latest end] time
      windows for servicing each node.
          - routes (list): A list of routes, where each route is a list of
      node IDs (integers, excluding the depot node 0).
    Return:
      bool: True if the solution satisfies the constraints, False
      otherwise.

    Note:
      The above "Args" cannot be added or modified, and no data outside
      the "solution" object should be used or added.
    """
    # Your code goes here...

    return True
```

**Function template for the Constraint Scorer:**

```python
def calculate_violation_score(solution: VrpState) -> float:
    """
    Args:
      solution (VrpState): An object representing the VRP solution.
        - problem_data (dict): A dictionary with problem details; Use only
        defined values in "Check Constraints Code".
        - routes (list): A list of routes, where each route is a list of
      node IDs (integers, excluding the depot).
        - cost (Callable[[], float]): Implemented method to compute and
      return the total cost of the solution's routes.
    Returns:
      violation_score (float): The calculated constraint violation score,
      where higher scores indicate more severe violations.

    Note:
      Do not use data or methods that is not contained in the "solution"
      object.
    """
    # Your code goes here ...
```

## D   VRP VARIANTS

Common problems in VRP are typically constructed based on a set of fundamental constraints, such as vehicle capacity, distance limits, time windows, pickup and delivery, same vehicle, and priority. These problems are widely used to evaluate the performance of multi-task algorithms (Jiang et al., 2025; Liu et al., 2024a; Li et al., 2024). By combining six representative constraints, excluding cases where vehicle capacity conflicts with pickup and delivery, a subset of 48 common problems is generated, as shown in Table 5.

Table 5: The 48 common problems constructed by six representative constraints.

| Problems | Vehicle Capacity | Distance Limit | Time Windows | Pickup and Delivery | Same Vehicle | Priority |
|---|---|---|---|---|---|---|
| VRP | | | | | | |
| PVRP | | | | | | ✓ |
| VRPS | | | | | ✓ | |
| PVRPS | | | | | ✓ | ✓ |
| VRPPD | | | | ✓ | | |
| PVRPPD | | | | ✓ | | ✓ |
| VRPPDS | | | | ✓ | ✓ | |
| PVRPPDS | | | | ✓ | ✓ | ✓ |
| VRPTW | | | ✓ | | | |
| PVRPTW | | | ✓ | | | ✓ |
| VRPSTW | | | ✓ | | ✓ | |
| PVRPSTW | | | ✓ | | ✓ | ✓ |
| VRPPDTW | | | ✓ | ✓ | | |
| PVRPPDTW | | | ✓ | ✓ | | ✓ |
| VRPPDSTW | | | ✓ | ✓ | ✓ | |
| PVRPPDSTW | | | ✓ | ✓ | ✓ | ✓ |
| VRPL | | ✓ | | | | |
| PVRPL | | ✓ | | | | ✓ |
| VRPLS | | ✓ | | | ✓ | |
| PVRPLS | | ✓ | | | ✓ | ✓ |
| VRPPDL | | ✓ | | ✓ | | |
| PVRPPDL | | ✓ | | ✓ | | ✓ |
| VRPPDLS | | ✓ | | ✓ | ✓ | |
| PVRPPDLS | | ✓ | | ✓ | ✓ | ✓ |
| VRPLTW | | ✓ | ✓ | | | |
| PVRPLTW | | ✓ | ✓ | | | ✓ |
| VRPLSTW | | ✓ | ✓ | | ✓ | |
| PVRPLSTW | | ✓ | ✓ | | ✓ | ✓ |
| VRPPDLTW | | ✓ | ✓ | ✓ | | |
| PVRPPDLTW | | ✓ | ✓ | ✓ | | ✓ |
| VRPPDLSTW | | ✓ | ✓ | ✓ | ✓ | |
| PVRPPDLSTW | | ✓ | ✓ | ✓ | ✓ | ✓ |
| CVRP | ✓ | | | | | |
| PCVRP | ✓ | | | | | ✓ |
| CVRPS | ✓ | | | | ✓ | |
| PCVRPS | ✓ | | | | ✓ | ✓ |
| CVRPTW | ✓ | | ✓ | | | |
| PCVRPTW | ✓ | | ✓ | | | ✓ |
| CVRPSTW | ✓ | | ✓ | | ✓ | |
| PCVRPSTW | ✓ | | ✓ | | ✓ | ✓ |
| CVRPL | ✓ | ✓ | | | | |
| PCVRPL | ✓ | ✓ | | | | ✓ |
| CVRPLS | ✓ | ✓ | | | ✓ | |
| PCVRPLS | ✓ | ✓ | | | ✓ | ✓ |
| CVRPLTW | ✓ | ✓ | ✓ | | | |
| PCVRPLTW | ✓ | ✓ | ✓ | | | ✓ |
| CVRPLSTW | ✓ | ✓ | ✓ | | ✓ | |
| PCVRPLSTW | ✓ | ✓ | ✓ | | ✓ | ✓ |

Table 6: Examples of constraint descriptions for 24 VRP variants.

| Problems | Problem Example |
|---|---|
| CVRP | The total load of each route must not exceed the vehicle capacity. |
| HCVRP | The total load of each route must not exceed the vehicle capacity. Specifically, there should be at least 3 routes where the total load is less than 100 units. |
| MCVRP | The total load of each route must not exceed the vehicle capacity. Additionally, nodes [12, 14] require deliveries of [70, 80] units of a new type of goods. The maximum load capacity for this type of goods on each route is 100 units, and problem data excludes information about new goods. |
| DCVRP | The total load of each route must not exceed the vehicle capacity. Specifically, for node [19], its base demand is augmented by 5 times the square root of the accumulated travel distance from the depot [0] to that node. |
| VRPL | Each route must not exceed 150 units in length. |
| HVRPL | Each route must not exceed 200 units in length, and at least three routes must have a total length of less than 150 units. |
| RVRP | After visiting node [17], the vehicle's remaining allowable travel distance for that route is reset to 150 units. At each node, the remaining driving distance cannot be negative. |
| DVRPL | Each route must not exceed 200 units in length. The vehicle's remaining range decreases with each visit. After visiting node [17], the remaining range will be halved. |
| VRPTW | The arrival time at each node must meet its specified time window. |
| HVRPTW | The arrival time at each node must meet its specified time window. Specifically, one route must have its start time is 300, while all other routes start with time 0. |
| VRPMTW | The arrival time at each node must meet its specified time window. For node [4], in addition to its original time window, an additional time window of [900, 950] is also available. |
| DVRPTW | The arrival time at each node must meet its specified time window. For node [18], the service time dynamically increases by the amount of time from the start of its time window to the arrival time. |
| VRPMPD | At Node [24], 10 units of goods are picked up. Unlike delivery, visiting a pickup node reduces the vehicle's available capacity. |
| HVRPMPD | At Node [24], 10 units of goods are picked up. Unlike delivery, visiting a pickup node reduces the vehicle's available capacity. Specifically, there should be at least 3 routes where the total load is less than 100 units. |
| MVRPMPD | Nodes [12, 14] require deliveries of [70, 80] units of a new type of goods. At Node [24], I pick up 20 units of these goods and 10 units of original goods. Before pick up, it needs to check whether sufficient goods have been delivered. Both types of goods are stored separately, with a maximum load of 100 units for new goods on each route, and problem data excludes information about new goods. |
| DVRPMPD | At Node [24], 10 units of goods are picked up, along with an additional amount calculated as 5 times the square root of the accumulated travel distance from the depot [0] to this node. Unlike delivery, visiting a pickup node reduces the vehicle's available capacity. |
| VRPSVC | Nodes [13, 23] must be on the same route. |
| CluVRP | Nodes [7, 10] must be on the same route, and these nodes must be visited consecutively. |
| VRPSO | Nodes [13, 23] must be on the same route, and node [13] must be visited before node [23]. |
| VRPILC | Nodes [7, 8] must not be on the same route. |
| PVRP | Nodes [5, 7] are priority points. |
| VRPRP | Node [8] is the priority node and must be one of the first three positions in at least one route. |
| VRPMP | Nodes [7, 5, 3] are priority nodes with strictly decreasing priority levels: [7] (highest), [5], and [3] (lowest). Higher-priority nodes must be visited before lower-priority ones and other nodes. |
| VRP-dRP | Nodes [7, 5, 3] follow the d-relaxed priority rule with decreasing priority: [7] (highest), [5], and [3] (lowest). Each node can be visited within its level or one level later, but no lower-priority node can be visited more than one level early. Other nodes are non-priority. |

# E   EXPERIMENTAL DETAILS

**Experimental environment:** Experiments are performed on a computer with an Intel Xeon Gold 6248R Processor (3.00 GHz), 128 GB system memory, and Windows 10.

## E.1   ANALYSIS OF ARS IN ROUTBENCH

In RoutBench, the ARS heatmaps illustrate the frequency of simultaneous errors encountered when solving composite Vehicle Routing Problems (VRPs). The horizontal and vertical axes correspond to 24 specific VRPs, with each cell representing the total number of errors occurring when solving a composite problem that includes both the row and column problems. The diagonal values indicate the total number of errors for individual problems, reflecting their inherent difficulty.

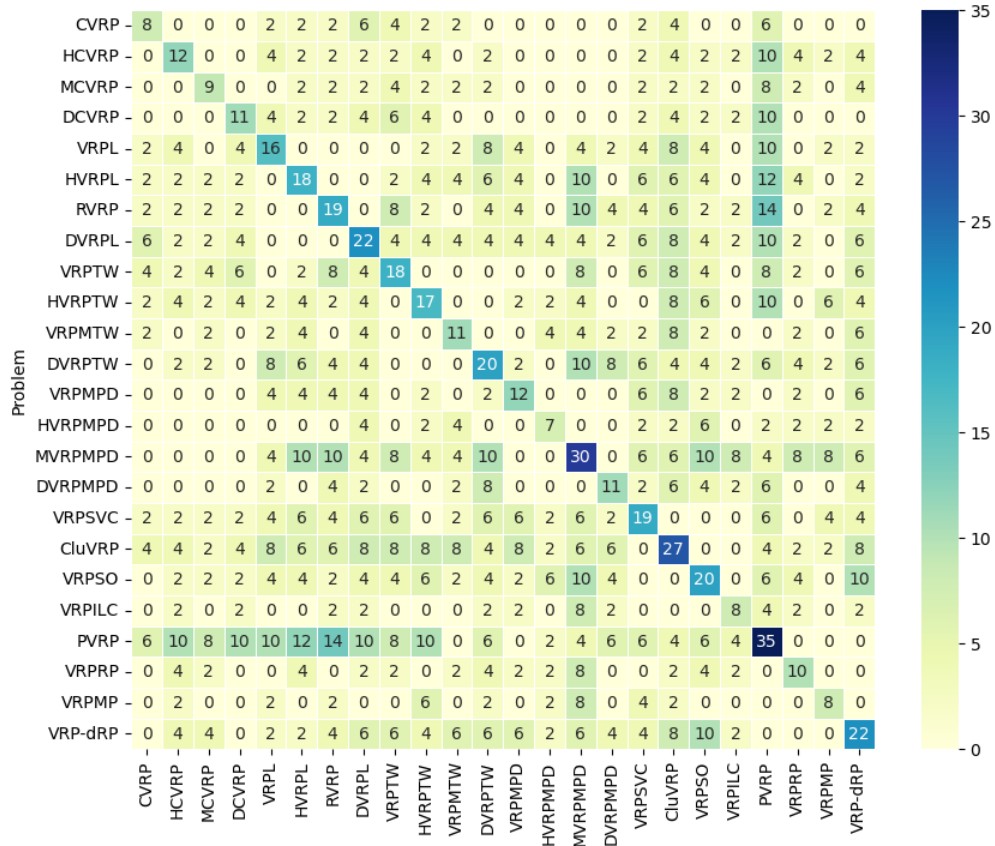

Figure 7: The heatmap of ARS within RoutBench-S shows the number of times errors occur simultaneously for the corresponding row (horizontal axis) and column (vertical axis). The diagonal values represent the number of errors for each individual problem.

In the RoutBench-S, the number of errors in priority problems is significantly higher than in other types of problems. This may be due to the inability of LLMs to adequately understand and handle priority issues. When transitioning to the RoutBench-H, the four time-window-related problems exhibit a significantly higher number of errors compared to other problem types. This suggests that time-window problems are inherently more complex. In contrast, regardless of whether the problems are RoutBench-S or RoutBench-H, our algorithm performs exceptionally well in terms of modeling success rates for capacity constraints and return point constraints.

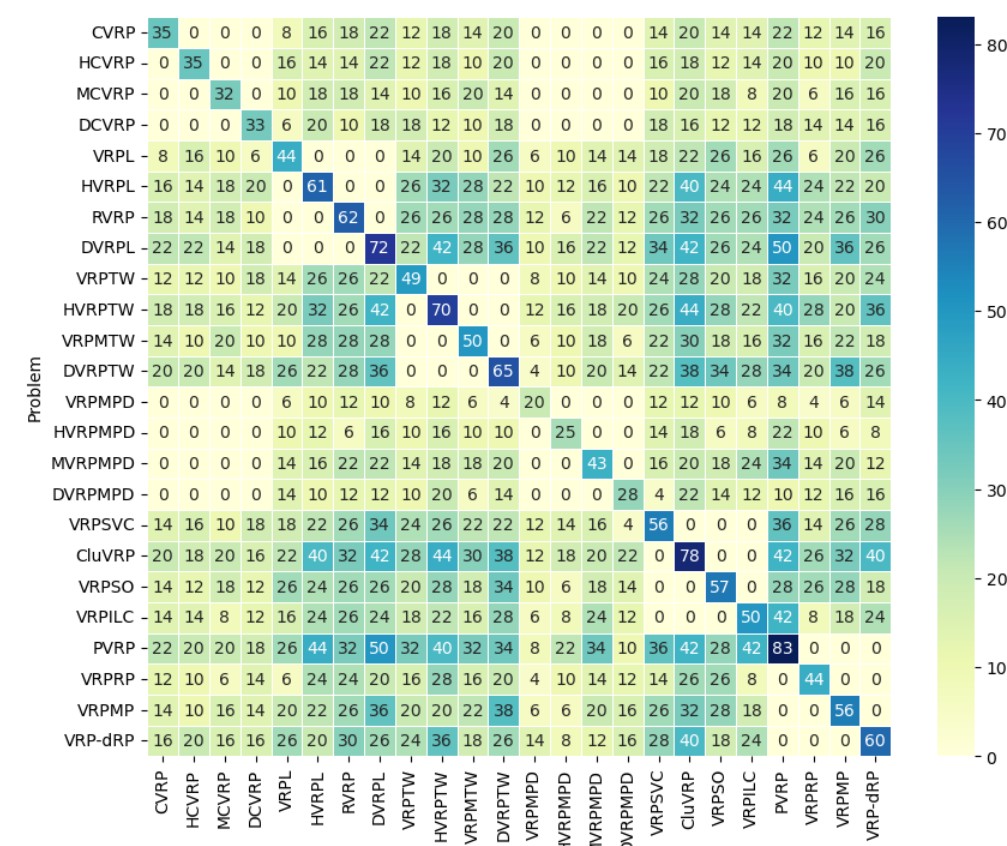

Figure 8: The heatmap of ARS within RoutBench-H shows the number of times errors occur simultaneously for the corresponding row (horizontal axis) and column (vertical axis). The diagonal values represent the number of errors for each individual problem.

### E.2    THE NUMBER OF LLM-SUGGESTED CONSTRAINTS

To better understand the relationship between the real constraint count in VRPs and the constraints suggested by the LLM agent, we analyze how the LLM adapts its recommendations based on the problem's complexity. As shown in Table 7, the data reveals a clear trend where the number of LLM-suggested constraints increases as the real constraint count grows. This suggests that the LLM agent effectively adapts its recommendations based on the complexity of the problem. Interestingly, the LLM agent tends to suggest slightly more constraints than the actual count, likely as a precautionary measure to ensure no potentially relevant constraints are overlooked.

Table 7: Analysis of the relationship between real constraint counts and LLM-suggested constraints in various VRPs, including the total number of problems analyzed for each constraint count and the corresponding average number of suggested constraints by the LLM agent.

| Constraint Count | Number of Problems | Average of LLM-Suggested Constraints |
|---|---|---|
| 1 | 8 | 1.25 |
| 2 | 92 | 2.43 |
| 3 | 400 | 3.81 |
| 4 | 263 | 4.84 |
| 5 | 237 | 5.63 |

### E.3   PROBLEM SET FOR TEST INSTANCES

As shown in Table 8, two common problems (e.g., CVRP and CVRPTW) and two dynamic problems (e.g., DCVRP and DCVRP-L) are presented, along with their respective settings and constraints. In this paper, "N" represents the number of nodes, "C" represents the vehicle capacity, and "L" represents the length of the maximum travel distance.

Table 8: Detailed problem set and constraints for test instances, including common and dynamic VRP variants.

| Problems | Setting | Problem Description |
|---|---|---|
| CVRP | C=200 | The total load of each route must not exceed the vehicle capacity. |
| CVRPTW | C=200 | The total load of each route must not exceed the vehicle capacity. The arrival time at each node must meet its specified time window. |
| DCVRP | C=200 | The total load of each route must not exceed the vehicle capacity. Specifically, for node [19], its base demand is augmented by 5 times the square root of the accumulated travel distance from the depot [0] to that node. |
| DCVRP-L | C=200, L=150 | The total load of each route must not exceed the vehicle capacity. Specifically, for node [19], its base demand is augmented by 5 times the square root of the accumulated travel distance from the depot [0] to that node. Each route must not exceed 150 units in length. |

### E.4   ENHANCING ARS WITH OTHER METHODS

Our framework, ARS, is designed to make it easier for LLMs to generate correct code for a wide range of VRPs, even when using foundation models. To demonstrate its potential and flexibility, we show that the performance of ARS can be further enhanced in two main ways: by integrating it with advanced prompting techniques and by leveraging more capable code generation models.

First, we explore improving ARS by incorporating established prompting techniques. Specifically, we enhance it with two methods: Reflexion (Shinn et al., 2024) and Self-debug (Chen et al., 2023). Table 9 presents a detailed comparison between the original ARS and these enhanced variants on the RoutBench benchmark. The results, obtained using the DeepSeek-V3 model, show that both enhancements lead to a clear improvement in success rate (SR) while simultaneously reducing the runtime error rate (RER). Notably, ARS+Self-debug achieves the highest success rates and the lowest error rates, demonstrating its effectiveness in refining the program generation process.

Table 9: Performance comparison of ARS and its enhanced variants (ARS+Reflexion and ARS+Self-debug) on RoutBench. The best results among these methods are highlighted in grey.

| Methods | RoutBench-S | | RoutBench-H | |
|---|---|---|---|---|
| | SR ↑ | RER ↓ | SR ↑ | RER ↓ |
| ARS+Reflexion | 78.20% | 0.20% | 63.20% | 2.00% |
| ARS+Self-debug | 78.80% | 0.00% | 68.80% | 0.60% |
| ARS | 77.20% | 2.80% | 61.60% | 5.60% |

Second, we evaluate the performance of ARS when using a more powerful code generation model. As shown in Table 10, we compare the results from the baseline DeepSeek-V3 model with those from Claude 3 Sonnet. The experiment shows that using a more advanced model provides a significant performance boost, achieving a much higher success rate on both the simple and hard tasks in RoutBench. These findings suggest that our framework is poised to improve further as language models and prompting techniques continue to advance.

Table 10: The performance of ARS is evaluated on more capable code generation models. The best results among these methods are highlighted in grey.

| | RoutBench-S | | RoutBench-H | |
|---|---|---|---|---|
| | SR ↑ | RER ↓ | SR ↑ | RER ↓ |
| DeepSeek-V3 | 77.20% | 2.80% | 61.60% | 5.60% |
| Claude-Sonnet-4-Thinking | 85.80% | 2.20% | 70.00% | 6.40% |

### E.5 STABILITY AND RELIABILITY OF ARS

To verify the stability and reliability of our framework, we evaluate its performance from two perspectives: consistency across multiple independent runs and robustness under different LLM parameters.

First, we conduct three independent runs of ARS on the RoutBench benchmark using the DeepSeek V3 model. The detailed results are presented in Table 11, which reports the success rates (SR) and runtime error rates (RER) for each trial. The results show consistent performance across the three runs, with low standard deviations in both SR and RER, indicating that ARS performs stably and reliably across different runs and benchmarks.

Table 11: Detailed results of three independent runs of ARS on RoutBench, including success rates, runtime error rates, averages, and standard deviations to evaluate stability and reliability.

| Experiment | RoutBench-S | | RoutBench-H | |
|---|---|---|---|---|
| | SR ↑ | RER ↓ | SR ↑ | RER ↓ |
| Run 1 | 77.20% | 2.80% | 61.60% | 5.60% |
| Run 2 | 80.60% | 2.00% | 64.20% | 2.60% |
| Run 3 | 76.40% | 2.80% | 64.80% | 5.80% |
| Average | 78.07% | 2.53% | 63.53% | 4.67% |
| Standard Deviation | 3.56% | 0.14% | 2.14% | 2.89% |

Second, we examine how ARS performs with various LLM parameters for the DeepSeek-V3. As shown in Table 12, we test different settings for temperature (T) and top-p. The results demonstrate that ARS maintains strong and robust performance across these configurations. This suggests that our framework is not overly sensitive to hyperparameter adjustments and can be effectively used with a variety of common settings.

Table 12: Performance of ARS on the RoutBench benchmark with different model parameters. The results for various settings of temperature (T) and top-p are presented to evaluate the reliability of this framework.

| Setting | RoutBench-S | | RoutBench-H | |
|---|---|---|---|---|
| | SR ↑ | RER ↓ | SR ↑ | RER ↓ |
| T = 0.3, Top P = 0.8 | 82.60% | 0.60% | 73.40% | 0.60% |
| T = 0.7, Top P = 1 | 77.20% | 2.80% | 61.60% | 5.60% |
| T = 1, Top P = 1 | 81.00% | 0.40% | 69.60% | 1.80% |

### E.6 BEST-KNOWN SOLUTIONS FOR ROUTBENCH

The best-known solutions (BKS) can be accessed through the provided link to the RoutBench repository. We provide the BKS for all instances in RoutBench, a comprehensive benchmark that encompasses 1,000 VRP variants with varying problem sizes (25, 50, and 100 nodes). For each instance, the BKS is obtained using ARS, which applies the correct Constraint-Aware Heuristic to ensure feasibility and solution quality. The algorithm is executed under rigorous stopping criteria: it

terminates when no improvement is observed for 1,000 consecutive generations or when the runtime exceeds one hour.

### E.7 DETAILS OF THE ABLATION STUDY

To further understand the contribution of each component in ARS, we conduct an ablation study using four different LLMs on common problems. The results are detailed in Table 13.

Table 13: Ablation study of the performance of ARS across four different LLMs. The best results among these methods are highlighted in grey.

| Methods | GPT-3.5-Turbo | GPT-4o | DeepSeek-V3 | LLaMA-3.1-70B |
|---|---|---|---|---|
| w/o Constraint Selection | 43.75% | 62.50% | 91.67% | 72.92% |
| w/o Database | 41.67% | 41.67% | 43.75% | 41.67% |
| ARS (full) | 47.92% | 91.67% | 95.83% | 77.08% |

First, removing the database (w/o Database) causes a sharp performance drop across all models. Without access to constraint examples, the LLMs struggle to generate correct code. Second, the impact of removing the constraint selection module (w/o Constraint Selection) varies by model. The effect is minimal on a less capable model like GPT-3.5-Turbo but significant for stronger ones like GPT-4o. Notably, DeepSeek-V3 remains robust to this change, showcasing its ability to handle unfiltered information. Overall, these results confirm that the full ARS framework, which uses both the database and constraint selection, achieves the best performance across all four LLMs.

### E.8 AN IN-DEPTH ANALYSIS OF FAILURE TYPES

We conducted an in-depth failure analysis. Even when a generated program is free of runtime errors, it can still produce an incorrect solution due to other, more subtle issues. Our analysis was performed on the unsuccessful programs generated by DeepSeek-V3 for the RoutBench benchmark.

The results of our analysis are detailed in Table 14. We found that the overwhelming majority of failures are caused by logical bugs, which were present in over 80% of the failed cases for both RoutBench-S and RoutBench-H. The second most frequent issue was incorrect constraint handling, affecting roughly 25% of the failures. In contrast, runtime errors were significantly less common, accounting for only about 10% of the issues. This clearly indicates that the main challenge is no longer just generating executable code, but ensuring the code is logically correct and properly adheres to all problem constraints. Notably, the RoutBench includes verification codes that validate whether the output produced by the generated program successfully satisfies all problem constraints.

Table 14: Distribution of failure types observed in programs generated for RoutBench-S and RoutBench-H, categorized into logical bugs, incorrect constraint handling, and runtime errors, with corresponding counts and rates.

| Failure Type | RoutBench-S | | RoutBench-H | |
|---|---|---|---|---|
| | Num. | Rate | Num. | Rate |
| Logical Bugs | 96 | 81.36% | 142 | 80.68% |
| Incorrect Constraint Handling | 32 | 27.12% | 47 | 26.70% |
| Runtime Error | 14 | 11.86% | 28 | 15.91% |
| Failure Problems | 118 | – | 176 | – |

### E.9 ANALYSIS OF END-TO-END RUNTIMES

To provide a complete picture of the latency of the framework, we measure the full end-to-end runtime, which includes the LLM inference time. We conduct three separate runs for each problem instance to ensure the results are reliable. Table 15 presents the detailed timings.

The results show that the LLM inference is a one-time cost that typically takes approximately 1 minute. After this initial code generation step, our framework solves the problem efficiently. On average, it finds solutions for 25, 50, and 100-node instances in about 5 seconds, 15 seconds, and under 5 minutes, respectively. Slight variations in these solving times can exist across different runs because the LLM generates the constraint-handling code. Overall, our framework offers two key advantages in terms of time:

1) **Faster Development Cycle:** The total time is significantly shorter compared to the traditional workflow, which often involves consulting experts and engaging in long development cycles for each new VRP variant.

2) **One-Time Cost:** Once the LLM-generated program is successful, it can be deployed across all problem instances of the same type without needing additional LLM inference.

It is also worth noting that our solver is developed in Python, while many other methods (e.g., OR-Tools) are written in C++. Python programs are generally 10 to 100 times slower than compiled C++ code. In the future, implementing the solver component in C++ can substantially reduce the computation time of our framework.

Table 15: Performance comparison of ARS execution and total end-to-end latency across various problem scales.

| Problems | Run | 25 Nodes | | 50 Nodes | | 100 Nodes | |
|---|---|---|---|---|---|---|---|
| | | Solver | End-to-End | Solver | End-to-End | Solver | End-to-End |
| CVRP | 1 | 2.3s | 41.7s | 19.1s | 70.3s | 2.4m | 3.3m |
| | 2 | 2.1s | 43.5s | 16.9s | 58.1s | 3.8m | 4.8m |
| | 3 | 2.7s | 42.8s | 17.4s | 92.7s | 3.3m | 4.3m |
| | Avg. | 2.4s | 42.7s | 17.8s | 73.7s | 3.1m | 4.1m |
| CVRPTW | 1 | 3.7s | 49.2s | 10.2s | 69.2s | 4.3m | 5.4m |
| | 2 | 2.5s | 48.5s | 15.4s | 50.6s | 3.9m | 5.1m |
| | 3 | 3.6s | 51.6s | 17.4s | 88.3s | 4.2m | 5.3m |
| | Avg. | 3.3s | 49.8s | 14.3s | 69.4s | 4.1m | 5.3m |

### E.10 EVALUATION OF ARS PERFORMANCE ON CVRPLIB

To further validate the effectiveness of ARS in solving real-world instances, we conducted experiments using five test suites from the CVRPLIB benchmark datasets. These datasets consist of 99 instances from Sets A, B, F, P, and X (Uchoa et al., 2017), encompassing graph scales ranging from 16 to 200 nodes, diverse node distributions, and varying customer demands.

For context, we show the results of state-of-the-art algorithms (HGS, LKH-3) and recent multi-task NCO methods (MTPOMO (Liu et al., 2024a), MVMoE (Zhou et al., 2024)), but these are for reference only. Notably, our approach focuses on enabling existing solvers to handle a wide range of VRPs, rather than claiming to achieve state-of-the-art (SOTA) results on specific problem types. As shown in Table 16, these specialized methods typically handle only a few dozen problem variants. Adapting them to a new problem often requires manual algorithm modifications and significant time. In contrast, our framework can handle over 1000 different VRP variants (as described in Appendix E.9) and can process a new variant in about one minute.

Table 16: The number of distinct VRP variants handled by ARS and other solving methods.

| | HGS | LKH-3 | MTPOMO | MVMoE | ARS |
|---|---|---|---|---|---|
| Num. of problems | 50+ | 53 | 11 | 16 | 1000 |

Table 17 presents a detailed performance comparison against these methods and three general-purpose solvers (CPLEX, OR-Tools, and Gurobi). As the table shows, ARS achieves the best performance among the four general solvers, demonstrating its competitive performance in solving these challenging instances.

Table 17: Performance comparison of ARS with other methods on CVRPLIB. The best results among the four solvers (i.e., CPLEX, OR-Tools, Gurobi, and ARS) are highlighted in grey.

| | Opt. | HGS | | LKH-3 | | MTPOMO | | MVMoE | |
| --- | --- | --- | --- | --- | --- | --- | --- | --- | --- |
| | | Obj. | Gap | Obj. | Gap | Obj. | Gap | Obj. | Gap |
| Set A | 1041.9 | 1042.2 | 0.00% | 1041.9 | 0.00% | 1087.9 | 5.07% | 1071.3 | 3.07% |
| Set B | 963.7 | 964.5 | 0.00% | 963.7 | 0.00% | 1006.9 | 4.86% | 999.2 | 3.94% |
| Set F | 707.7 | 709 | 0.00% | 707.7 | 0.00% | 820 | 16.23% | 791.3 | 12.16% |
| Set P | 587.4 | 586.9 | 0.00% | 587.4 | 0.00% | 629.3 | 11.10% | 614 | 6.76% |
| Set X | 27220.1 | 27223.7 | 0.01% | 27281.4 | 0.02% | 28952.5 | 6.09% | 28688.4 | 5.19% |
| Avg. | 6104.2 | 6105.3 | 0.00% | 6116.4 | 0.00% | 6499.3 | 8.67% | 6432.8 | 6.22% |
| | Opt. | CPLEX | | OR-Tools | | Gurobi | | ARS | |
| | | Obj. | Gap | Obj. | Gap | Obj. | Gap | Obj. | Gap |
| Set A | 1041.9 | 1096.5 | 5.24% | 1058.9 | 1.63% | 1067.3 | 2.44% | 1055.5 | 1.31% |
| Set B | 963.7 | 1003.6 | 4.14% | 973.3 | 1.00% | 990.9 | 2.82% | 973 | 0.96% |
| Set F | 707.7 | 789.3 | 11.53% | 728.7 | 2.97% | 728.7 | 2.97% | 727 | 2.73% |
| Set P | 587.4 | 612.5 | 4.27% | 592 | 0.78% | 594.9 | 1.28% | 591.1 | 0.62% |
| Set X | 27220.1 | 32044.1 | 17.72% | 28209.6 | 3.64% | 28977.7 | 6.46% | 28142.4 | 3.39% |
| Avg. | 6104.2 | 7109.2 | 8.58% | 6312.5 | 2.00% | 6471.9 | 3.19% | 6297.8 | 1.80% |

## F  EXAMPLES OF SOLVER CODES

To better analyze different methods for solving VRPs, we provide code examples for four approaches: our solver, Gurobi, OR-Tools, and CPLEX. As a case study, these methods are applied to the Capacitated Vehicle Routing Problem with Time Windows (CVRPTW) to illustrate their respective requirements and complexities.

**Our Solver Code:**

```python
def check_constraints(solution: VrpState) -> bool:
    """
    Args:
      solution (VrpState): An object representing the VRP solution.
        - problem_data (dict): A dictionary with problem details:
            - "edge_weight": A 2D NumPy array of node distances.
            - "demand": A 1D NumPy array of node demands (0 for the depot).
            - "capacity": Maximum load capacity per vehicle.
            - "service_time": A 1D NumPy array of node service times (0 for
        the depot).
            - "time_window": A list of [earliest start, latest end] time
        windows for servicing each node.
          - routes (list): A list of routes, where each route is a list of
        node IDs (integers, excluding the depot node 0).

    Return:
      bool: True if the solution satisfies the constraints, False
        otherwise.

    Note:
      The above "Args" cannot be added or modified, and no data outside
        the "solution" object should be used or added.
    """
    for route in solution.routes:
        # Check Vehicle Capacity Constraint
        total_demand = sum(solution.problem_data["demand"][node] for node in
         route)
        if total_demand > solution.problem_data["capacity"]:
            return False

        # Check Time Windows Constraint
        current_time = 0  # Start at time 0
        tour = [0] + route + [0]  # Add depot at the beginning and end of
        the route

        for idx in range(1, len(tour)):
            arrive_time = current_time + solution.problem_data['edge_weight'][
        tour[idx - 1]][tour[idx]]
            wait_time = max(0, solution.problem_data['time_window'][tour[idx
        ]][0] - arrive_time)  # Wait if early
            current_time = arrive_time + wait_time

            tw_start, tw_end = solution.problem_data['time_window'][tour[idx]]
            if current_time > tw_end:
                return False

            # Add the service time for the current node after arriving and
        waiting
            current_time += solution.problem_data['service_time'][tour[idx]]

    return True
```

**Gurobi Code:**

```python
from gurobipy import Model, GRB, quicksum
from typing import List

def find_feasible_routes(solution: VrpState) -> List[List[int]]:
    """
    Finds feasible routes for a CVRPTW using an optimization solver.

    Args:
      solution (VrpState): An object representing the VRP solution.
        - problem_data (dict): A dictionary with problem details:
            - "demand": A 1D NumPy array of node demands (0 for the depot).
            - "vehicles": Maximum vehicle.
            - "capacity": Maximum load capacity per vehicle.
            - "edge_weight": A 2D NumPy array or matrix of distances
      between nodes.

    Return:
      routes (list): A list of optimized routes, where each route is a
      list of node IDs (integers, excluding the depot node 0).
    """
    data = solution.problem_data
    num_nodes = len(data["demand"])
    demand = data["demand"]
    capacity = data["capacity"]
    vehicles = data["vehicles"]
    edge_weight = data["edge_weight"]
    service_time = data["service_time"]
    time_window = data["time_window"]

    model = Model("CVRPTW_PathLength")
    model.setParam('TimeLimit', 50)

    # Decision Variables
    x = model.addVars(num_nodes, num_nodes, vtype=GRB.BINARY, name="x")
    u = model.addVars(num_nodes, vtype=GRB.CONTINUOUS, name="u")  # For
      capacity
    arrival_time = model.addVars(num_nodes, vtype=GRB.CONTINUOUS, lb=0,
      name="arrival_time")

    # Objective: Minimize total distance
    model.setObjective(
      quicksum(edge_weight[i][j] * x[i, j] for i in range(num_nodes) for j
       in range(num_nodes) if i != j),
      GRB.MINIMIZE
    )

    # Core Constraints
    model.addConstrs((quicksum(x[i, j] for i in range(num_nodes) if i != j
      ) == 1 for j in range(1, num_nodes)),
                     "in_degree")
    model.addConstrs((quicksum(x[i, j] for j in range(num_nodes) if i != j
      ) == 1 for i in range(1, num_nodes)),
                     "out_degree")
```

```
47   model.addConstr(quicksum(x[0, j] for j in range(1, num_nodes)) <=
       vehicles, "max_vehicles")
48   model.addConstr(quicksum(x[0, j] for j in range(1, num_nodes)) ==
       quicksum(x[j, 0] for j in range(1, num_nodes)),
49                   "depot_flow")
50
51   # MTZ Capacity Constraints
52   model.addConstr(u[0] == 0, "u_depot")
53   for j in range(1, num_nodes):
54     model.addConstr(u[j] >= demand[j], f"u_min_{j}")
55     model.addConstr(u[j] <= capacity, f"u_max_{j}")
56     for i in range(num_nodes):
57       if i != j:
58         model.addConstr(
59           u[j] >= u[i] + demand[j] - capacity * (1 - x[i, j]),
60           f"mtz_cap_{i}_{j}"
61         )
62
63   # Time window constraints
64   model.addConstr(arrival_time[0] == 0, name="StartTimeDepot")
65   model.addConstrs(
66     (arrival_time[i] + service_time[i] + edge_weight[i][j] <=
       arrival_time[j] + (1 - x[i, j]) * 1e6
67      for i in range(num_nodes) for j in range(1, num_nodes) if i != j),
       name="TravelTime")
68   model.addConstrs((arrival_time[j] >= time_window[j][0] for j in range
       (1, num_nodes)), name="EarliestArrival")
69   model.addConstrs((arrival_time[j] <= time_window[j][1] for j in range(
       num_nodes)), name="LatestArrival")
70
71   # Solve
72   model.optimize()
73
74   # Core route extraction logic
75   routes = []
76   if hasattr(model, 'status') and model.status in [GRB.OPTIMAL, GRB.
       TIME_LIMIT]:
77     visited = set()
78     for j in range(1, num_nodes):  # Skip depot
79       if x[0, j].x > 0.5 and j not in visited:
80         current, route = j, []
81         while current != 0:  # Until return to depot
82           route.append(current)
83           visited.add(current)
84           current = next((k for k in range(num_nodes) if k != current
       and x[current, k].x > 0.5), 0)
85         if route:
86           routes.append(route)
87   return routes
```

From the provided code examples, it is clear that our solver requires the least amount of code to be generated by an LLM. This is attributed to the simplicity and flexibility of our method, which avoids the need for verbose or overly rigid programming constructs. In contrast, the code examples for Gurobi, OR-Tools, and CPLEX are more complex, primarily due to their reliance on strict syntax rules and detailed configurations.

**OR-Tools Code:**

```python
def find_feasible_routes(solution: VrpState) -> list[list[int]]:
    """
    Finds feasible routes for a VRP problem using an optimization solver.

    Args:
      solution (VrpState): An object representing the VRP solution.
        - problem_data (dict): A dictionary with problem details:
            - "capacity": Maximum load capacity per vehicle.
            - "demand": 1D numpy array, demand [0]=0.
            - "time_window": A 2D NumPy array where each row represents the
      [earliest, latest] time windows for servicing each node.
            - "service_time": service time for each node.
            - "edge_weight": 2D distance matrix.
    Return:
      routes (list): A list of optimized routes, where each route is a
      list of node IDs (integers, excluding the depot node 0).

    Note:
      The above "Args" cannot be added or modified, and no data outside
      the "solution" object should be used or added.
      Ensure each node is visited exactly once.
    """
    # Data Preprocessing
    data = solution.problem_data
    # Ensure all data types are correct
    edge_weight = data['edge_weight'].astype(np.int64)
    demands = [int(x) for x in data['demand'].astype(np.int64)]
    time_windows = [[int(tw[0]), int(tw[1])] for tw in data['time_window'
      ]]
    service_times = [int(st) for st in data['service_time'].astype(np.
      int64)]
    vehicle_capacity = int(data['capacity'])
    num_nodes = len(data["demand"])
    num_vehicles = num_nodes - 1

    # Create the routing index manager
    manager = pywrapcp.RoutingIndexManager(num_nodes, num_vehicles, 0)
    # Create Routing Model
    routing = pywrapcp.RoutingModel(manager)

    # Create and register a transit callback for distance.
    def distance_callback(from_index, to_index):
        # Returns the distance between the two nodes.
        from_node = manager.IndexToNode(from_index)
        to_node = manager.IndexToNode(to_index)
        return edge_weight[from_node][to_node]

    transit_callback_index = routing.RegisterTransitCallback(
      distance_callback)

    # Define cost of each arc.
    routing.SetArcCostEvaluatorOfAllVehicles(transit_callback_index)

    # Add Capacity constraint
    def demand_callback(from_index):
```

```
50      # Returns the demand of the node.
51      from_node = manager.IndexToNode(from_index)
52      return demands[from_node]
53
54   demand_callback_index = routing.RegisterUnaryTransitCallback(
       demand_callback)
55   routing.AddDimensionWithVehicleCapacity(
56      demand_callback_index, 0, [vehicle_capacity] * num_vehicles, True, '
       Capacity'
57   )
58
59   # Add Time Window constraint
60   def time_callback(from_index, to_index):
61      # Returns the travel time between the two nodes (excluding service
       time).
62      from_node = manager.IndexToNode(from_index)
63      to_node = manager.IndexToNode(to_index)
64      return edge_weight[from_node][to_node]  # Only travel time, no
       service time
65
66   time_callback_index = routing.RegisterTransitCallback(time_callback)
67   horizon = time_windows[0][1]  # Maximum time limit for routes (depot's
        latest end time)
68   routing.AddDimension(
69      time_callback_index, horizon, horizon, False, 'Time'
70   )
71
72   # Set time windows for all nodes (including the depot)
73   time_dimension = routing.GetDimensionOrDie('Time')
74   for node_index in range(num_nodes):  # Include depot node
75      index = manager.NodeToIndex(node_index)
76      time_dimension.CumulVar(index).SetRange(time_windows[node_index][0],
        time_windows[node_index][1])
77      # Add service time to the departure time
78      routing.solver().Add(
79         time_dimension.CumulVar(index) + service_times[node_index] <=
       time_dimension.CumulVar(index) + service_times[node_index]
80      )
81
82   # Solver Configuration
83   search_params = pywrapcp.DefaultRoutingSearchParameters()
84   search_params.first_solution_strategy = (
85      routing_enums_pb2.FirstSolutionStrategy.PATH_CHEAPEST_ARC
86   )
87   search_params.time_limit.seconds = 50  # Timeout protection
88   # Route Extraction
89   routes = []
90   if (result := routing.SolveWithParameters(search_params)):
91      for vehicle_id in range(routing.vehicles()):
92         index = routing.Start(vehicle_id)
93
94         route = []
95         while not routing.IsEnd(index):
96            node = manager.IndexToNode(index)
97            if node != 0:  # Filter out depot node
98               route.append(node)
99            index = result.Value(routing.NextVar(index))
100        if route:  # Filter out empty routes
101           routes.append(route)
102
103   return routes
```

**CPLEX Code:**

```python
from docplex.mp.model import Model

def find_feasible_routes(solution: VrpState) -> list[list[int]]:
    """
    Finds feasible routes for a CVRPTW using CPLEX/DOcplex.

    Args:
      solution (VrpState): Contains problem_data dict with keys:
        - "capacity": Q
        - "demand": 1D numpy array, demand[0]=0
        - "edge_weight": 2D distance matrix
        - "time_window": time window, time_window[i]=(earliest, latest)
        - "service_time": service time for each node
        - optionally "num_vehicles": K
    Returns:
      List of routes (each a list of customer indices, excluding depot 0).
    """
    data = solution.problem_data
    Q = data["capacity"]
    demand = data["demand"]
    edge_weight = data["edge_weight"]
    time_window = data["time_window"]
    service_time = data["service_time"]
    num_nodes = len(demand)
    V = list(range(num_nodes))
    N = V[1:]
    K = data.get("num_vehicles", None)

    # Get the earliest and latest times for time windows
    earliest = {{i: time_window[i][0] for i in V}}
    latest = {{i: time_window[i][1] for i in V}}

    # Calculate a large number M as the upper bound for time constraints
    M = sum(latest[i] + service_time[i] + max(edge_weight[i]) for i in V)

    # Build model
    model = Model(name="VRPTW")
    model.parameters.timelimit = {run_time}  # Set time limit for solving

    # Decision variables x[i,j]
    arcs = [(i, j) for i in V for j in V if i != j]
    x = model.binary_var_dict(arcs, name="x")

    # MTZ load variables u[i] with lb=demand[i], ub=Q
    u = model.continuous_var_dict(
        N,
        lb={{i: float(demand[i]) for i in N}},
        ub=Q,
        name="u"
    )

    # Time variables: arrival time at node i
    t = model.continuous_var_dict(V, lb=0, name="t")

    # Objective: minimize total distance
    model.minimize(model.sum(edge_weight[i, j] * x[i, j] for i, j in arcs)
      )
```

```python
57   # Degree constraints: each customer exactly one in and one out
58   for i in N:
59     model.add_constraint(model.sum(x[i, j] for j in V if j != i) == 1)
60     model.add_constraint(model.sum(x[j, i] for j in V if j != i) == 1)
61
62   # Vehicle count limit at depot
63   if K is not None:
64     model.add_constraint(model.sum(x[0, j] for j in N) <= K)
65     model.add_constraint(model.sum(x[i, 0] for i in N) <= K)
66
67   # MTZ sub-tour elimination / capacity constraints
68   for i, j in arcs:
69     if i != 0 and j != 0:
70       model.add_constraint(u[i] + demand[j] <= u[j] + Q * (1 - x[i, j]))
71
72   # Time window constraints
73   # Enforce time window ranges
74   for i in V:
75     model.add_constraint(earliest[i] <= t[i])
76     model.add_constraint(t[i] <= latest[i])
77
78   # Ensure time logic along the route is correct
79   for i, j in arcs:
80     if j != 0:  # No need to consider time window for returning to the
         depot
81       model.add_constraint(
82         t[i] + service_time[i] + edge_weight[i, j] <= t[j] + M * (1 - x[
         i, j])
83       )
84
85   # Solve
86   sol = model.solve(log_output=True)
87   if sol is None:
88     logger.error("No solution found.")
89     return []
90
91   # Extract used arcs
92   used = [(i, j) for (i, j) in arcs if x[i, j].solution_value > 0.5]
93
94   # Build successor map
95   succ = {{i: j for i, j in used}}
96
97   # Reconstruct routes from depot
98   routes = []
99   starts = [j for i, j in used if i == 0]
100  for st in starts:
101    route = []
102    cur = st
103    while cur != 0:
104      route.append(cur)
105      cur = succ.get(cur, 0)
106    routes.append(route)
107
108  return routes
```

## G  POTENTIAL SOCIETAL IMPACT

Automatic Routing Solver (ARS) has the potential to drive significant societal advancements by addressing complex real-world Vehicle Routing Problems (VRPs) in industries such as logistics, transportation, and healthcare. ARS leverages Large Language Model (LLM) agents to automate the design of constraint-aware heuristic solvers, offering flexibility and efficiency in solving diverse VRP scenarios.

ARS brings two main strengths to VRP solutions: 1) it dynamically adapts to diverse practical constraints, providing robust solutions without requiring extensive manual design, and 2) it introduces interpretability, enabling decision-makers to better understand and customize routing solutions for specific needs. These features make ARS a valuable tool for optimizing operations, reducing costs, and improving resource utilization across sectors. Additionally, the RoutBench benchmark ensures rigorous evaluation of ARS, further validating its real-world applicability.

However, ARS is not without challenges. Over-reliance on automated solvers may limit human oversight, and misinterpretation of results could lead to suboptimal decisions. Furthermore, the use of LLMs in ARS raises concerns about data security, as sensitive operational or constraint information could inadvertently be exposed. Addressing these risks will be crucial to ensuring ARS's safe and ethical deployment. Lastly, while ARS demonstrates strong performance, its success rate may vary depending on the complexity of constraints, potentially delaying decision-making in highly intricate scenarios.

## H  USE OF LLMS

In this work, we utilize Large Language Models (LLMs) for two primary purposes. First, we use an LLM as a writing assistant to help refine sentence structure, improve clarity, and correct grammatical errors throughout the manuscript. Moreover, the LLM plays a crucial role in our experimental methodology. Specifically, we employ the LLM to automatically generate code for different solving frameworks, enabling them to handle a wide range of VRP variants.

## I  LICENSES

The licenses and URLs of the baseline methods are provided in Table 18.

Table 18: A summary of licenses.

| Resources | Type | URL | License |
|---|---|---|---|
| CPLEX | Code | https://www.ibm.com/products/ilog-cplex-optimization-studio | Available for academic research use |
| OR-Tools | Code | https://github.com/google/or-tools | Apache License 2.0 |
| Gurobi | Code | https://www.gurobi.com/ | Available for academic research use |
| LKH-3 | Code | http://webhotel4.ruc.dk/~keld/research/LKH-3/ | Available for academic research use |
| PyVRP-HGS | Code | https://github.com/PyVRP/PyVRP | MIT License |
| Reflexion | Code | https://github.com/noahshinn/reflexion | MIT License |
| PHP | Code | https://github.com/chuanyang-Zheng/Progressive-Hint | Available online |
| CoE | Code | https://github.com/xzymustbexzy/Chain-of-Experts/tree/main | Available online |
| Self-verification | Code | https://github.com/Zhehui-Huang/LLM_Routing | MIT License |
| Solomon | Dataset | http://vrp.atd-lab.inf.puc-rio.br/index.php/en/ | Available for academic research use |
| CVRPLib | Dataset | http://vrp.atd-lab.inf.puc-rio.br/ | Available for academic research use |

