# OpenReview forum: "ARS: Automatic Routing Solver with Large Language Models"
_ICLR.cc/2026/Conference — Submitted to ICLR 2026_

### Official Review · Reviewer_mt1a · 2025-10-28

**Soundness:** 2
**Presentation:** 2
**Contribution:** 2
**Rating:** 4
**Confidence:** 4

**Summary:**

This paper presents ARS (Automatic Routing Solver), a framework that leverages LLMs to solve real-world VRPs with complex and diverse constraints. Unlike traditional solvers, ARS can automatically generate constraint-aware heuristics from natural language descriptions, reducing the need for manual expert modeling as problem constraints become varied or more complicated.​

The contributions include:  1) The method uses LLMs to interpret users’ descriptions of routing problems, selects relevant constraints from a database, and generates Python code that is plugged into a backbone heuristic solver. This allows ARS to flexibly adapt to many different and realistic VRP scenarios, from heterogeneous vehicle fleets to time windows, dynamic demands, and priority rules.​
2) The authors introduce RoutBench, a comprehensive benchmark suite with 1,000 VRP variants built from combinations of 24 practical constraints. This allows for systematic and rigorous testing of automatic routing solvers in diverse, real-world cases.​
3) ARS achieves high success rates (over 90% on standard VRPs and over 60% on complex RoutBench problems), outperforming seven other LLM-based code generation methods and three general-purpose solvers. The approach also makes it easier for LLMs to generate correct code, with less complexity and higher efficiency.​
4) By automating constraint handler generation and linking it to established search heuristics, ARS shows strong generalization across problem types and is ready for extension beyond routing, such as bin packing.​
5) ARS automatically generates constraint-aware heuristics to solve diverse real-world VRP variants, using LLMs and a large, systematic benchmark for evaluation.

**Strengths:**

1. ARS uses Large Language Models to automatically create heuristics and constraint handlers for many real-world routing problems. This makes solver setup fast and easy, helping users adapt to changing or complex requirements without needing expert coding or modeling.​
2. The RoutBench benchmark covers 1,000 problem variants built from 24 real-world constraints. ARS demonstrates strong generalization and robust performance across this diverse set, including highly constrained and previously untested scenarios.​
3. On standard VRPs and complex RoutBench problems, ARS achieves much higher success rates and code correctness (over 90% on common tasks, above 60% on tough cases), outperforming both 7 alternative LLM-based approaches and 3 established solvers in efficiency and reliability.​
4. By generating only the parts of code needed for constraint handling, ARS significantly lowers the amount of generated code required and reduces errors compared to general-purpose modeling frameworks.​
5. The framework works with various state-of-the-art language models (GPT-4, DeepSeek, LLaMA), showing improved results as model quality increases and supporting scalability for different environments.​

**Weaknesses:**

1. While ARS claims to automate solver design with LLMs, the underlying routing backbone relies heavily on conventional metaheuristics (destroy/repair, 2-OPT, SWAP, SHIFT), offering little methodological advancement over established VRP solvers. The core innovation is largely in combining existing components rather than proposing fundamentally new search or optimization strategies.​

2. The framework delegates constraint handling and validation almost entirely to LLM agents, making solution accuracy and feasibility highly dependent on the ability of language models to interpret and synthesize correct code from often ambiguous or poorly-specified natural language descriptions. This introduces significant risk in reliability, especially as constraints grow complex or nuanced beyond the LLM’s training distribution.​

3. The paper fails to seriously investigate the risk of hallucination, syntax errors, and subtle bugs common in LLM-generated code. Runtime error rates are measured, but no robust fail-safe is proposed for erroneous validation or constraint logic. This potentially impacting real-world deployment.​

4. The backbone search algorithm is fixed and non-adaptive, restricting ARS to a single solution-generation paradigm. This limits potential improvements from modern neural combinatorial optimization, hybrid strategies, or domain-specific deep learning advances that could outperform classic heuristics, especially for harder VRP variants and scalability.​

5. RoutBench is much richer than previous benchmarks but is entirely built from synthetic combinations of constraint types with fixed datasets and rules. Real-world VRPs often include messy, dynamic, multi-modal data, and less formalized or evolving constraints. These are not examined, and thus claims about real-world generalization are partly unproven.​

6. The approach shows success up to 200-node problems, but does not evaluate extreme scalability to industrial-size problems (thousands of locations, live routing updates), nor does it rigorously compare wall-time performance to state-of-the-art solvers like HGS or commercial systems in diverse operational contexts.​

7. ARS focuses on constraints expressible in the current framework and retrievable from its database. Constraints that require joint probabilistic modeling, dynamic uncertainties, learning-based preference handling, or integration with external platforms are out of scope.​

8. Automated code generation for optimization raises serious issues in operational safety, security, and fairness that are only mentioned briefly. There is no systematic audit or mitigation for risks of unintended or adversarial LLM outputs, data leakage, or regulatory compliance.​

9. While the paper claims extensibility to other domains (e.g., bin packing), there is no empirical evidence or adaptation strategies for handling fundamentally different combinatorial problems, which limits generality.

**Questions:**

1. Considering that the proposed framework relies on LLMs to generate constraint-checking code from natural language problem descriptions, how the authors guarantee correctness, robustness, and safety of the generated code? This is more important in the presence of ambiguous, under-specified, or adversarial input. What evidence can the authors provide that ARS would not silently accept incorrect solutions or propagate bugs to mission-critical logistics environments?​

2. The proposed benchmark (RoutBench) is well-constructed but fully synthetic. How does ARS handle the noisy, unstructured, and evolving constraint types found in real operational datasets, such as time-varying networks, legal constraints, or incomplete business rules? Can authors demonstrate successful generalization or present negative results for deployment in high-stakes, dynamic, real-world logistics problems?​

3. While the authors claim that ARS can be extended to other combinatorial domains (bin packing, job-shop scheduling, etc.), the experiments are strictly for VRPs. What architectural or algorithmic changes are needed for ARS to work on these different optimization classes, and what unique challenges arise if the problem structure diverges (e.g., from graph to sequence to set)?

---

> ### Author Response · Authors · 2025-11-25
>
> Dear Reviewer mt1a,
>
> Thank you for recognizing that ARS makes solver setup fast and easy for complex routing problems and for highlighting its robust performance across diverse scenarios. We have addressed each point below:
>
>
> **W1. Originality**
>
> Thank you for your feedback on the originality of our work. Our main contributions are twofold:
>
> 1. **New paradigm for automation:** We propose a framework that enables a search algorithm to handle a wide range of VRPs. The core design lies not in the specific search operators, but in **a new paradigm that makes it much easier for an LLM to generate correct code compared to other general-purpose solvers** (as shown in Figure 4).
>
> 2. **Systematic benchmark:** We introduce RoutBench, a benchmark with 1,000 VRP variants. **This design not only expands the test scale by two orders of magnitude but also provides an opportunity to evaluate algorithms on unseen VRPs.**
>
> **Our core innovation introduces a novel framework.** We show how to flexibly and rapidly adapt a heuristic algorithm to solve a class of problems based on user natural language descriptions, unlike other general-purpose solvers that require highly standardized modeling to solve new problems.
>
> As mentioned in Section 6, building upon existing methods allows us to better highlight our framework. This approach demonstrates **its potential to be combined with other heuristic algorithms and generalized to other domains**, as we show below with the **3D bin packing problem**.
>
>
> **W2. Dependent on the ability of LLMs**
>
> Thank you for raising this important concern. While using LLMs to understand natural language descriptions and synthesize correct code for optimization problems is a common research direction [1-3], we agree that relying on LLMs for code generation introduces reliability risks. **Our goal is to improve development efficiency, not to completely replace human developers**. To that end, we have considered three main aspects:
>
> 1. **Rigorous reliability validation:** We introduced RoutBench, a benchmark with 1,000 VRP variants, most of which are new or unseen. This allows us to specifically test reliability when constraints become complex or fall outside the LLM's training distribution.
>
> 2. **Improving reliability:** Our framework is designed to make it much easier for an LLM to generate correct code compared to general-purpose solvers (as shown in Figure 4). By simplifying the task for the LLM, we inherently improve the reliability of the output.
>
> 3. **User Validation:** Our method is designed to assist users, significantly reducing development costs from months to minutes. However, when a problem is described using ambiguous or poorly-specified natural language, **the resulting code can contain bugs, whether it is written by a human expert or an LLM**.
>
> Therefore, **for any problem requiring high reliability, we state that users must validate the generated algorithm (e.g., by testing with small-scale instances) before deploying it in critical real-world scenarios**.

---

> ### Author Response · Authors · 2025-11-25
>
> **W3 & Q1. Risks of LLM-Generated Code**
>
> We agree with your concerns about the risks of LLM-generated code. We considered this concern from three perspectives:
>
> 1. **Analyzing the Causes of Unreliability**. In Table 14, we analyzed the different types of errors: **Logical Bugs, Incorrect Constraint Handling, and Runtime Errors**. We found that the main reason for these errors was Logical Bugs, which are typically caused by LLM hallucinations.
>
> 2. **Improving Reliability**. As shown in Table 9, **we combined ARS with other LLM-based methods** (e.g., reflection and self-debugging) to improve reliability. This allowed the system to successfully and automatically handle over 70% of the problems in RoutBench.
>
>     **Our proposed framework is designed to make it much easier for an LLM to generate correct code compared to other general-purpose solvers**. In the future, ARS can be combined with more effective strategies to further increase the reliability of the LLM’s output.
>
>
> 3. **Real-World Deployment. Our method is designed to significantly reduce development costs by assisting users with unseen problems, not to fully replace human experts**. A user without specialized programming skills can provide their problem requirements to ARS and receive a customized algorithm in about a minute. This offers clear practical value compared to hiring a modeling team for several months.
>
>     However, when a problem is described in natural language, especially in the presence of ambiguous, under-specified, or adversarial input, **the resulting code may contain bugs. This is true whether the code is written by a human expert or generated by an LLM**.
>
> Therefore, **for any problem that requires high reliability, users must validate the generated algorithm (for example, by testing it with small-scale instances) before deploying it in critical real-world scenarios**.
>
> **W4. Backbone search algorithm**
>
> Thank you for your insightful suggestion about the potential for more adaptive search algorithms.
>
> The primary goal of this work is to clearly demonstrate our proposed paradigm, which enables an LLM to flexibly adapt an existing algorithm to a new class of optimization problems. By keeping the backbone algorithm fixed, we can better highlight how our framework makes it much easier for an LLM to generate correct code compared to other general-purpose solvers.
>
> We agree that exploring different types of algorithms is a valuable direction for future work. We plan to integrate ARS with more advanced methods to further improve its search capabilities and apply it to other problem domains. As a preliminary example, we show an application of this paradigm to the 3D Bin Packing problem below.

---

> ### Author Response · Authors · 2025-11-25
>
> **W5. RoutBench**
>
> Thank you for your feedback regarding RoutBench. We would like to clarify its design and contribution.
>
> 1. **Sourcing from Real-World Constraints:** All constraint types in RoutBench are sourced from real-world problems described in academic literature. As detailed in Table 1, **these include not only common constraints like capacity and time windows but also more complex scenarios such as heterogeneous fleets and dynamic loads**.
>
> 2. **The Significance of RoutBench:** RoutBench is the first benchmark to include over 1,000 VRP variants. **This design not only expands the test scale by two orders of magnitude but also provides an opportunity to evaluate algorithms on unseen VRPs**.
>
> RoutBench is intended to serve as a standardized and scalable benchmark for the community. It is designed to be extensible, and **we have already expanded it to include more complex problem types, such as VRP variants with conflicting constraints caused by unreasonable user demands, resulting in no feasible solution**.
>
> > I need to make sure each route is no longer than 0 units.
>
>
> **W6. Performance**
>
> Thank you for raising this point. While our framework is not inherently limited by problem size, we will leave this for future work. For example, ARS can be combined with more advanced methods, which would allow it not only to solve a wide range of variants but also to achieve more competitive search performance on larger-scale problems.
>
> **Live routing updates**. As shown in Table 3, **ARS achieved the best results** when compared to other general-purpose solvers on multiple types of dynamic VRPs.
>
> **Reduced development time**. Developing a specialized solver for a particular constraint is often **a significant effort requiring expert knowledge**. For example, when handling a dynamic VRP, an expert with specialized knowledge might spend several days using advanced standardized modeling techniques to create the constraint-handling code for a general-purpose solver, even with an LLM's assistance.
>
> In contrast, **the design of the ARS framework makes it much easier for an LLM to generate correct code compared to other general-purpose solvers**. Therefore, **it can directly understand a problem described in natural language by a user and automatically handle the new constraint within a minute**. This capability massively reduces development costs.
>
> **W7. The Scope of Constraints ARS Can Handle**
>
> Thank you for your feedback. We would like to clarify that ARS can handle a wide variety of constraints, far exceeding the six examples stored in our database.
>
> 1. **Diversity of Constraints:** ARS can handle a wide variety of constraints. For instance, RoutBench contains 24 different types of real-world constraints, all of which ARS can process. These include not only common constraints like capacity and time windows, but also **more complex scenarios such as heterogeneous fleets and dynamic loads**.
>
> 2. **The Role of the Database:** The purpose of the database in ARS is to serve as a few-shot reference for the LLM agent, helping it effectively use code templates. This process is designed to make it easier for the agent to handle new and unseen constraints, which go far beyond the ones in the database.
>
> 3. **Handling Other Constraint Types:** In principle, **as long as a constraint can be verified with code, it can be handled by ARS. This is because the foundation of our method is to use a checker to determine if a given solution satisfies the constraints**. This means that the constraints you mentioned, such as those involving dynamic uncertainties or integration with external platforms, are within the scope of our framework, provided they can be verified with code.
>
> **W8. Safety and Fairness of Automated Code Generation**
>
> We acknowledge your concerns about the safety and fairness of automated code generation. We agree that these are critical issues and wish to clarify the intended role and responsible use of our method.
>
> 1. **Real-World Deployment:** Our method is designed to significantly **reduce development costs by assisting users with unseen problems, not to fully replace human experts**. A user without specialized programming skills can provide their problem requirements to ARS and receive a customized algorithm in about a minute. This offers clear practical value compared to hiring a modeling team for several months.
>
> 2. **Human Oversight:** We agree that for any problem requiring systematic audits or risk mitigation, the resulting code must be carefully reviewed. It is important to note that any code may contain bugs, regardless of whether it is written by a human expert or generated by an LLM.
>
> Therefore, **for any problem that requires high reliability, users must validate the generated algorithm (for example, by testing it with small-scale instances) before deploying it in critical real-world scenarios**.

---

> > ### Author Response · Authors · 2025-11-25
> >
> > **W9 & Q3. Other COPs**
> >
> > This paper focuses mainly on the VRP. However, the framework is generalizable to other optimization problems, and **we have already begun applying it to the 3D Bin Packing Problem (BPP)**.
> >
> > Specifically, we use a classic packing algorithm as the backbone and test the framework on a diverse set of 128 BPP variants that incorporate 10 different types of constraints.
> >
> > |Method|SR|
> > |-|-|
> > |Standard Prompting|46.09%|
> > |Automatic Packing Solver|80.47%|
> >
> >
> > The results show that the Automatic Packing Solver (APS) successfully handled 80.47\% of these packing problems. This strongly suggests that our framework is **not limited to VRP and can be adapted for other COPs**.
> >
> > As for adapting this framework to a broader range of problems and algorithms, we will leave this for future work. If this paper is accepted, we are committed to open-sourcing code and providing corresponding tutorials. We hope that this framework can be applied to other classes of optimization problems, such as scheduling and beyond.
> >
> > **Q2. Handling Real-World Constraints**
> >
> > Thank you for this question regarding the real-world applicability of ARS.
> >
> > **The Design of RoutBench**. All constraint types in RoutBench are sourced from real-world problems described in academic literature. As detailed in Table 1, these include not only common constraints like capacity and time windows but also **more complex scenarios such as heterogeneous fleets and dynamic loads**.
> >
> >
> > **Handling Other Constraint Types**. **As long as a constraint can be verified with code, it can be handled by ARS**. This is because **the foundation of our method is to use a checker to determine if a given solution satisfies the constraints**. This means that the constraints you mentioned, such as those involving dynamic uncertainties or evolving business rules, are within the scope of our framework, provided they can be verified with code.
> >
> > **Negative Results and Infeasible Problems**. Your question about deployment in high-stakes, dynamic problems is a crucial one. A common real-world scenario is **when user demands are unreasonable, leading to conflicting or overly strict constraints. By definition, such problems have no feasible solution**.
> >
> > In these cases, ARS still generates constraint-handling code based on the problem description. Consequently, during the search phase, the algorithm will be unable to find a feasible solution. However, the scoring function will continuously guide the search operators to minimize the degree of constraint violation.
> >
> > Finally, when a stopping condition (such as the time limit) is met, ARS outputs an infeasible solution with the minimum possible degree of violation. In other words, **this approach provides a practical solution that satisfies the requirements as much as possible**.
> >
> > Thank you for your thoughtful consideration of how ARS would handle various real-world and extreme scenarios. We hope our responses help to address your concerns.
> >
> > [1] Chain-of-experts: When llms meet complex operations research problems. ICLR, 2023.
> >
> > [2] DRoC: Elevating large language models for complex vehicle routing via decomposed retrieval of constraints. ICLR, 2025.
> >
> > [3] ORLM: Training Large Language Models for Optimization Modeling. Operations Research, 2025.

---

> > > ### Comment · Reviewer_mt1a · 2025-11-26
> > > **Clarifies framework; core concerns remain.**
> > >
> > > Thank you for your detailed and technical response. It clarifies the intended contribution (LLM-assisted constraint routing paradigm plus RoutBench), provides useful reliability/error analyses, and offers preliminary evidence of extensibility, while some of my concerns about safety guarantees, large-scale realism, and deployment in high‑stakes settings remain only partially addressed.

---

> ### Author Response · Authors · 2025-12-01
>
> Dear Reviewer mt1a,
>
> We are glad our response clarified our core contributions regarding the LLM-assisted paradigm, RoutBench, and our analyses on reliability and extensibility. Our further response is as follows:
>
> **Q1. Clarification on safety in high-stakes settings**
>
> Thank you for your concern about the practical application of ARS. However, as we have explained above, we want to clarify the intended role and responsible use of our method.
>
> 1. **Real-World Deployment:** Our method is designed to significantly **reduce development costs by assisting users with unseen problems, not to fully replace human experts**. A user without specialized programming skills can provide their problem requirements to ARS and receive a customized algorithm in about a minute. This offers clear practical value compared to hiring a modeling team for several months.
>
> 2. **Human Oversight:** We agree that for any problem requiring systematic audits or risk mitigation, the resulting code must be carefully reviewed. It is important to note that **any code may contain bugs, regardless of whether it is written by a human expert or generated by an LLM**.
>
> Therefore, **for any problem that requires high reliability, users must validate the generated algorithm (for example, by testing it with small-scale instances) before deploying it in critical real-world scenarios**.
>
>
>
> **Q2. Large-scale realism**
>
> On large-scale problems, our results show that ARS outperforms specialized methods on **instances with over 4,000 nodes**.
>
> The search performance of ARS is provided by the underlying search algorithm. In the future, we will further improve search performance by incorporating more efficient search algorithms.
>
> | Instance | Scale | BQ (Gap) | LEHD RRC1000 (Gap) | ARS (Gap) |
> | :--- | :--- |:---------|:-----------|:----------|
> | Leuven1 | 3,000 | 15.39%   | 10.71%     | **7.66%**     |
> | Leuven2 | 4,000 | 25.69%   | 21.22%     | **17.14%**    |
>
> We hope our responses help to address your concerns.

---

### Official Review · Reviewer_BaF6 · 2025-10-29

**Soundness:** 2
**Presentation:** 3
**Contribution:** 2
**Rating:** 4
**Confidence:** 4

**Summary:**

This paper introduces the Automatic Routing Solver (ARS), a novel framework that leverages Large Language Models (LLMs) to automatically generate solvers for a wide variety of Vehicle Routing Problems (VRPs). The core idea is to use an LLM not to create an entire solving algorithm from scratch, but to generate problem-specific, constraint-aware heuristic code that can be integrated into a robust, general-purpose metaheuristic backbone. The ARS framework consists of a database of fundamental VRP constraints, an LLM-driven module that selects relevant constraints and generates new "checker" and "scorer" programs based on a natural language problem description, and an augmented heuristic solver that uses this generated code to guide its search.

As a second major contribution, the paper presents RoutBench, a new comprehensive benchmark comprising 1,000 VRP variants derived from 24 different real-world constraints. This benchmark is designed to rigorously evaluate the generalization capabilities of VRP solvers. Experiments show that ARS significantly outperforms other LLM-based methods in successfully generating correct code and provides a more effective and efficient framework for tackling diverse VRPs compared to general-purpose commercial solvers like Gurobi and CPLEX when paired with an LLM.

**Strengths:**

1. The paper's primary strength is the innovative design of the ARS framework, which intelligently separates the general solver backbone from the LLM-generated, problem-specific heuristic components. This is a clever and effective way to combine the reasoning power of LLMs with the proven search capabilities of metaheuristics.

2. The introduction of RoutBench is a major contribution in its own right. It provides a large-scale, diverse, and well-structured testbed for evaluating the generalization capabilities of VRP solvers, which has been lacking in the field.

3. The work addresses a highly significant and practical problem: reducing the immense manual, expert-driven effort required to design and implement solvers for the vast and growing number of VRP variants encountered in logistics and transportation.

**Weaknesses:**

1. While the paper proposes the ARS framework, its originality is limited. The framework's RAG component utilizes existing technology, the "checker" and "scorer" steps are based on established ideas from heuristic VRP solvers, and the subsequent heuristic algorithm is also a pre-existing method. Overall, the lack of substantial novel content is the paper's most significant weakness.

2. The paper relies on a single-point-based search framework. It is unclear how the LLM-generated Constraint-Aware Heuristic (CAH) would perform when integrated with other powerful metaheuristic backbones, such as population-based genetic algorithms or ant colony optimization.

3. While ARS is shown to be superior to standard prompting, the prompts provided in the appendix appear carefully engineered. The framework's robustness to minor variations in prompt phrasing or in the natural language problem descriptions is not explored.

**Questions:**

1. The manuscript would benefit from a careful proofread to correct typos and imprecise descriptions. For example:
On line 49, "Gurubi" is misspelled and should be "Gurobi". On line 409, the text discussing solver performance refers to "Table 1" when it should be "Table 3". On line 322, the claim that the distribution in RoutBench "reflects the proportions of the full set of 5624 problems" is not entirely accurate, as the visual distributions in Figure 2 show noticeable differences. On line 372, the statement that other methods have a success rate "merely around 10%" on RoutBench-H is imprecise. The actual results in Table 2 range from 10.8% to 15.6%. A thorough review of the entire text is recommended to find and fix any similar issues.

2. The caption for Table 3 states, "The table presents the gaps compared to the results obtained by ARS." Shouldn't the gap be compared to the BKS (Best Known Solutions)? Or, were the BKS in this paper actually the results obtained by ARS? Please provide an explanation.

3. The paper rightly states it does not aim for state-of-the-art (SOTA) performance on specific VRPs. However, the performance gap between ARS and specialized solvers like HGS (Table 17) is large. A more detailed discussion of this trade-off would be beneficial.

4. Could you provide a concrete example of a "Logical Bug" from your failure analysis (Table 14)? What does such a bug look like in the generated code, and what underlying reasoning failure from the LLM do you believe it represents?

---

> ### Author Response · Authors · 2025-11-25
>
> Dear Reviewer BaF6,
>
> Thank you for your insightful feedback. We are very encouraged that you recognized our work addresses a highly significant and practical problem and highlighted the contribution of RoutBench. We are happy to answer your questions.
>
> **W1. Novelty**
>
> Thank you for your feedback on the novelty of our work. Our main contributions are twofold:
>
> 1. **New paradigm for automation:** We propose a framework that enables a search algorithm to handle a wide range of VRPs. The core design lies not in the specific search operators, but in **a new paradigm that makes it much easier for an LLM to generate correct code compared to other general-purpose solvers** (as shown in Figure 4).
>
> 2. **Systematic benchmark:** We introduce RoutBench, a benchmark with 1,000 VRP variants. **This design not only expands the test scale by two orders of magnitude but also provides an opportunity to evaluate algorithms on unseen VRPs.**
>
> **Our core innovation is methodological.** We show **how to flexibly and rapidly adapt a heuristic algorithm to solve a class of problems based on user natural language descriptions**, unlike other general-purpose solvers that **require highly standardized modeling** to solve new problems.
>
> **W2. Integration with other metaheuristics**
>
> We agree that integrating the LLM-generated Constraint-Aware Heuristic (CAH) with other powerful metaheuristic backbones is an excellent and practically meaningful direction, one that could lead to further performance improvements.
> However, this paper's main focus is on generality rather than performance.
>
>
> We greatly appreciate the reviewer’s suggestion.
> We plan to explore this in our future work, as combining ARS with other powerful metaheuristics would allow it not only to solve a wide range of variants but also to achieve better search performance on them.

---

> > ### Author Response · Authors · 2025-11-25
> >
> > **W3. Sensitivity to Phrasing Changes**
> >
> > Thank you for your comment. We tested the framework's robustness using minor variations of the natural language descriptions for RoutBench. Our results show that **ARS is not sensitive to minor variations in phrasing**. Across these tests, the success rate changed by less than 5%.
> >
> >
> > |                  |RoutBench-S|       |RoutBench-H|        |
> > |------------------|----------------|-------|-|--------|
> > |                  |SR| RER   |SR| RER    |
> > | Minor Variations | 68.80% | 6.40% | 43.00% | 12.60% |
> > | Original         | 73.20% | 5.20% | 46.80% | 11.80% |
> >
> > In addition, we tested ARS on VRPs from IndustryOR [1], an industrial benchmark consisting of various real-world operations research problems, where each problem is presented as a natural language description of a real-world scenario.
> > For example, **ARS can solve one of the most difficult VRP variants in IndustryOR**, the CVRP with Hard Time Windows (CVRPHTW), from a natural language description.
> >
> > Below is the natural language description of the problem:
> >
> > ```
> > The Vehicle Routing Problem (VRP) was first proposed by Dantzig and Ramser in 1959. It is a classic combinatorial optimization problem. The basic VRP can be described as follows: in a certain area, there is a number of customers and a distribution center or depot ...
> >
> > The Vehicle Routing Problem with Time Windows (VRPTW) is a classic variant of the VRP. There are many real-world applications of VRPTW, as customer locations often have service time windows. For instance ...
> >
> > The Vehicle Routing Problem with Hard Time Windows (VRPHTW) can be described as follows: within a region, there is a set of customer locations and a central depot. Vehicles must start from the depot and return to the depot, following continuous paths ...
> >
> > Now we consider a major enterprise logistics provider, 'Global Logistics', is responsible for providing precise material delivery services for multiple high-end office buildings and shops in a city's central business district (CBD). Due to traffic control in the CBD and the specific receiving requirements of the customers, the delivery task is highly challenging.
> >
> > **Specific Requirements:**
> >
> > 1. **Delivery Task**: There are 20 customers requiring delivery service on the day, and the demands of all customers must be met.
> > 2. **Vehicle Constraints**: The company can use at most 5 trucks, and the capacity of each truck is 200 units.
> > 3. **Capacity Constraint**: The total demand of all customers on a single route must not exceed the truck's maximum capacity (200 units).
> > 4. **Time Window Constraint**: Each customer has a strict 'hard time window.' Service must begin within this specified time window. Early arrivals must wait, and late arrivals are not permitted.
> > 5. **Service Time**: Due to the complex handover procedures at customer sites, a fixed service time of 90 minutes is required for unloading, handover, and paperwork at each customer location.
> > 6. **Optimization Objective**: While satisfying all constraints, the company's objective is to **minimize the total distance traveled by all vehicles** to reduce operational costs.
> >
> > **Data Details:**
> >
> > * **Central Depot (Depot 0)**:
> > * Coordinates: (40, 50)
> > * Operating Time Window: [0, 1236] (minutes)
> > * **Customer Locations (Customers 1-20)**: The coordinates, demand, service time window, and service duration for each customer are shown in the table below.
> >
> > | Customer ID | Coordinates (X, Y) | Demand (units) | Time Window (minutes) | Service Duration (minutes) |
> > | :--- | :--- | :--- |:--- | :--- |
> > | 1 | (45, 68) | 10 | [912, 967] | 90 |
> > ...
> > | 20 | (30, 50) | 10 | [10, 73] | 90 |
> >
> > Now, please provide an operations research model for this VRPHTW.
> > ```
> >
> >
> > **Output from ARS**
> >
> > > Route #1: [13, 17, 18, 19, 15, 16, 14, 12]
> > >
> > > Route #2: [5, 3, 7, 8, 10, 11, 9, 6, 4, 2, 1]
> > >
> > > Route #3: [20]
> > >
> > > Objective Value: 175

---

> ### Author Response · Authors · 2025-11-25
>
> **Q1. Revisions for clarity and accuracy**
>
> Thank you for your meticulous feedback. We have performed a thorough review of the entire manuscript and have corrected the typos and imprecise descriptions.
>
> **Q2. Clarification on Gap Benchmark**
>
> Thank you for pointing this out. For all instances in Table 3, **ARS found the Best Known Solution (BKS)**. We will revise the caption to avoid misunderstanding.
>
>
> **Q3. Detailed discussion**
>
> Thank you for this insightful comment. We will add a more detailed discussion about the trade-off in generality and performance.
>
> We agree that highly efficient, specialized solvers like HGS and LKH3 achieve SOTA performance on several specific problems, as they have been developed over decades. However, **we think the performance gap (approximately 2% as shown in Table 17) is a worthwhile trade-off for the significant generality ARS provides**.
>
> **ARS allows a user without expert programming skills to describe their unique problem in natural language and receive a tailored solving algorithm in about a minute**. Compared to a specialized team spending weeks or months developing a custom algorithm, ARS significantly reduces the development cost and time, making it **a highly valuable tool in practical scenarios**.
>
> **This trade-off is particularly important in practical scenarios**. When a practitioner faces a new or previously unsolved problem, **the initial goal is often to obtain a feasible solution quickly, even if it is suboptimal, rather than spending significant time and resources to develop a specialized solver**. After all, without a method to solve the problem, performance is a moot point.
>
> Furthermore, **ARS is currently the only method capable of handling over 1,000 different VRP variants**. In contrast, **specialized solvers like HGS and LKH-3, despite decades of development, handle around 50 variants each (as shown in Table 16). ARS can solve two orders of magnitude more problem types than existing solvers**. Although a performance gap of around 2% exists for specific, well-studied problems compared to specialized solvers, we think the trade-off of performance for generality is worthwhile.
>
> Finally, we believe that in the future, combining ARS with other powerful metaheuristics would allow it not only to solve a wide range of variants but also to achieve more competitive search performance on them.
>
>
> **Q4. Logical bug example**
>
> Here is a simple example of a logical bug.
>
> > **Problem description**: Nodes [7, 8] must not be on the same route. Nodes [5, 7] are priority points.
>
> ```python
> def check_constraints(solution: VrpState) -> bool:
>
>     # Check that nodes 7 and 8 are on the same route
>     for route in solution.routes:
>         if 7 in route and 8 in route:
>             return False
>
>     # Check that nodes 5 and 7 are on the same route (priority points)
>     found_5 = False
>     found_7 = False
>     for route in solution.routes:
>         if 5 in route:
>             found_5 = True
>         if 7 in route:
>             found_7 = True
>     if not (found_5 and found_7):
>         return False
>
>     return True
> ```
>
>
> In this example, Node [7] is both a priority point and is also related to "same route constraint". The LLM might mistakenly apply "same route constraint“ to all priority points.
>
> We think **these errors are caused by LLM hallucinations**, as the risk of such mistakes increases with the amount of information the LLM must process.
>
> **Our framework helps to reduce this risk**. As shown in Figure 5, **ARS requires the LLM to generate the fewest lines of code compared to other general-purpose solvers**. This is because ARS allows the LLM to focus solely on generating constraint-handling code. In contrast, **other solvers require the LLM to take an entire framework (often hundreds of lines of code) as input, and then generate constraint-handling code within it that must fit into a highly standardized modeling structure**. This places **a much heavier burden on the LLM and increases the risk of logical errors**.
>
> We thank the reviewer for these helpful comments. We hope our responses have been helpful, and please feel free to let us know if you have any further questions, as we would appreciate the chance to respond again.
>
> [1] ORLM: Training Large Language Models for Optimization Modeling. Operations Research, 2025.

---

### Official Review · Reviewer_dTBv · 2025-10-29

**Soundness:** 3
**Presentation:** 2
**Contribution:** 2
**Rating:** 4
**Confidence:** 4

**Summary:**

This paper proposes an LLM-assisted framework that turns natural-language VRP into code for a constraint-aware heuristic. It retrieves exemplar constraints, generate a checker and violation scorer, and plugs them into a destroy–repair + local search backbone.

**Strengths:**

• The paper releases RoutBench: 1,000 VRP variants (each with NL description, data, and validation code), which represents a broad benchmark contribution.

• The ablations show each component of the framework matters, indicating the effectiveness of the design.

**Weaknesses:**

•	The evaluation is based on the correctness/coverage of the per-instance validation code. It is not reliable enough to ensure that the generated program works for a class of VRP. If a checker under-specifies edge cases, SR can be overstated.

•	Best-Known Solutions (BKS) for RoutBench are produced by ARS itself under strict stops. It seems that this method cannot ensure the actual (near)optimal solution, and thus leads to a benchmark circularity risk.

•	The superior performance partly reflect a competent destroy-repair + local search backbone. There’s limited comparison to stronger VRP metaheuristics (e.g., state-of-the-art HGS variants) within the same ARS-style interface, or introducing the LLM-based heuristic discovery methods.

•	The paper hints at portability to other COPs (e.g., 3D bin packing) but provides no experiments. There is no evidence that if ARS can serve as “general framework” also for other COPs.

**Questions:**

(1)	In Table 17, the comparative result between ARS and other baselines are reported. Can ARS always produce feasible solutions for the CVRPLIB instances? If not, the SR metric should also be indicated.

(2)	LLMs such as gpt-3.5-turbo are very old now. It would be better if new-generation LLMs, including reasoning model, can be evaluated in Figure 6.

(3)	Is that possible to use ARS to solve VRP described by any NL description? For example, a user may choose to describe a VRP by a real-world scenario.

---

> ### Author Response · Authors · 2025-11-25
>
> Dear Reviewer dTBv,
>
> Thank you for recognizing RoutBench as a broad benchmark contribution.
>  Below, we provide detailed responses to the your questions.
>
>
> **W1. SR can be overstated***
>
> Thank you for this insightful comment. We have reviewed the LLM-generated code and find that the **Success Rate (SR) is not overestimated**, for the following reasons:
>
> **Edge cases.** The generated code is free of simple edge case errors. For instance, when handling a boundary condition like the maximum route length, the code correctly uses
>
> ```python
> if route_route > max_route_length:
>     return false
> ```
>
> and avoids other mistakes, such as using >= instead.
>
> **Main type of errors.** As shown in Table 14, **the main source of errors is "logical errors," not edge cases**.
> For example, if a point is both a priority point and is also related to "same route constraint," the LLM might mistakenly apply "same route constraint“ to all priority points.
>
> **Testing with multiple instances.** To avoid overstating the SR, **we test each problem with instances of different sizes** (see the Supplementary Material for details).
>
> **W2. Best-Known Solutions**
>
> Thank you for raising this point. To handle 1,000+ VRP variants, using ARS is the most practical approach to establish the BKS for RoutBench.
>
>
> **Solving 1,000+ VRPs is difficult.** Finding the optimal solution for routing problems is incredibly challenging because they are NP-hard. This is particularly true for the problems in RoutBench, as they involve many complex rules like dynamic and non-linear constraints.
>
> **Existing methods typically handle only dozens of VRP variants**. For example:
>
> - LEHD [1] can only handle TSP and CVRP.
>
> - MTPOMO [2] can handle 11 VRP variants derived from 5 constraints.
>
> - CaDA [3] can handle 16 VRP variants derived from 5 constraints.
>
> **BKS for RoutBench. ARS is the only method we know of that can handle all 1,000+ VRP variants**, solving two orders of magnitude more problems than other existing methods. Therefore, to establish a baseline BKS for RoutBench, we used ARS and ran it for one hour on each instance.
>
> **Avoid the circularity risk.** To further validate the quality of ARS, **we also tested it on classic problems within RoutBench (like CVRP and CVRPTW) that other SOTA solvers can handle**. As shown in Table 3, the results demonstrate that ARS achieves the same BKS as these other advanced methods on these classic problems.
>
> **Act as a leaderboard.** Finding the true optimal solutions for these complex problems is difficult, so RoutBench can also act as a leaderboard, encouraging the research community to create better algorithms. As new and more powerful solvers are developed, they can challenge and improve the BKS in RoutBench.
>
> **W3. Other methods**
>
> Thank you for your suggestion. However, the main focus of our paper is on generality, rather than on solving efficiency.
>
> **Core contribution.** We aim to propose a general solver that automatically handles a wide range of VRPs, reducing the need for expert developers and lowering real-world costs.
>
> **Comparison with other methods.** In Table 16, we compare our method with **HGS variants, which can handle about 50 different VRP variants**. In contrast, **ARS can handle over 1,000 different VRP variants**.
>
> We also provide results for HGS on the classic problem (i.e., CVRP) in Table 17. We agree that specialized solvers like HGS and LKH3 achieve SOTA performance, as they have been developed over decades for these specific problems. We use these SOTA solvers as a reference and do not claim that our method can outperform these classic methods on such well-studied problems.
>
> **Combining with LLM-based methods.** We completely agree with you on this point. A promising direction for future work is to explore LLM-based automatic algorithm design (e.g., FunSearch [4] and EoH [5]) and combine it with our framework. This could automatically create highly effective algorithms for specific VRP variants or even particular problem instances.

---

> > ### Author Response · Authors · 2025-11-25
> >
> > **W4. Other COPs**
> >
> > This paper focuses mainly on the VRP. However, the framework is generalizable to other optimization problems, and **we have already begun applying it to the 3D Bin Packing Problem (BPP)**.
> >
> > Specifically, we use a classic packing algorithm as the backbone and test the framework on a diverse set of 128 BPP variants that incorporate 10 different types of constraints.
> >
> > |Method|SR|
> > |-|-|
> > |Standard Prompting|46.09%|
> > |Automatic Packing Solver|80.47%|
> >
> >
> > The results show that the Automatic Packing Solver (APS) successfully handled 80.47\% of these packing problems. This strongly suggests that our framework is **not limited to VRP and can be adapted for other COPs**.
> >
> > If this paper is accepted, we are committed to open-sourcing code and providing corresponding tutorials. We hope that this framework can be applied to other classes of optimization problems, such as scheduling and beyond.
> >
> > **Q1. Feasible solutions for CVRPLIB**
> >
> > Thank you for pointing this out. For the experiments in Table 17, **we provided ARS with the correct code** to analyze its solving efficiency. Therefore, ARS always generates a feasible solution for these instances. We will add a note to the paper to make this clear.
> >
> > **Q2. new-generation LLMs**
> >
> > Thinks. **We have added a new-generation model, Claude-Sonnet-4-Thinking**, which successfully handles over 85\% of RoutBench-S and over 70\% of RoutBench-H problems (as shown in Table 10). The results suggest that ARS has significant potential and is poised to improve further as language models continue to advance.

---

> > > ### Author Response · Authors · 2025-11-25
> > >
> > > **Q3. VRP described by any NL description**
> > >
> > > Yes, ARS can solve VRPs described in any natural language, **as long as these constraints can be verified by code**.
> > >
> > > To validate this, we tested ARS on VRPs from IndustryOR [6], the industrial benchmark consisting of various real-world operations research problems. For example, **ARS can solve one of the most difficult VRP variants in IndustryOR**, the CVRP with Hard Time Windows (CVRPHTW), from a natural language description.
> > >
> > > Below is the natural language description of the problem:
> > >
> > > ```
> > > The Vehicle Routing Problem (VRP) was first proposed by Dantzig and Ramser in 1959. It is a classic combinatorial optimization problem. The basic VRP can be described as follows: in a certain area, there is a number of customers and a distribution center or depot ...
> > >
> > > The Vehicle Routing Problem with Time Windows (VRPTW) is a classic variant of the VRP. There are many real-world applications of VRPTW, as customer locations often have service time windows. For instance ...
> > >
> > > The Vehicle Routing Problem with Hard Time Windows (VRPHTW) can be described as follows: within a region, there is a set of customer locations and a central depot. Vehicles must start from the depot and return to the depot, following continuous paths ...
> > >
> > > Now we consider a major enterprise logistics provider, 'Global Logistics', is responsible for providing precise material delivery services for multiple high-end office buildings and shops in a city's central business district (CBD). Due to traffic control in the CBD and the specific receiving requirements of the customers, the delivery task is highly challenging.
> > >
> > > **Specific Requirements:**
> > >
> > > 1. **Delivery Task**: There are 20 customers requiring delivery service on the day, and the demands of all customers must be met.
> > > 2. **Vehicle Constraints**: The company can use at most 5 trucks, and the capacity of each truck is 200 units.
> > > 3. **Capacity Constraint**: The total demand of all customers on a single route must not exceed the truck's maximum capacity (200 units).
> > > 4. **Time Window Constraint**: Each customer has a strict 'hard time window.' Service must begin within this specified time window. Early arrivals must wait, and late arrivals are not permitted.
> > > 5. **Service Time**: Due to the complex handover procedures at customer sites, a fixed service time of 90 minutes is required for unloading, handover, and paperwork at each customer location.
> > > 6. **Optimization Objective**: While satisfying all constraints, the company's objective is to **minimize the total distance traveled by all vehicles** to reduce operational costs.
> > >
> > > **Data Details:**
> > >
> > > * **Central Depot (Depot 0)**:
> > > * Coordinates: (40, 50)
> > > * Operating Time Window: [0, 1236] (minutes)
> > > * **Customer Locations (Customers 1-20)**: The coordinates, demand, service time window, and service duration for each customer are shown in the table below.
> > >
> > > | Customer ID | Coordinates (X, Y) | Demand (units) | Time Window (minutes) | Service Duration (minutes) |
> > > | :--- | :--- | :--- |:--- | :--- |
> > > | 1 | (45, 68) | 10 | [912, 967] | 90 |
> > > ...
> > > | 20 | (30, 50) | 10 | [10, 73] | 90 |
> > >
> > > Now, please provide an operations research model for this VRPHTW.
> > > ```
> > >
> > >
> > > **Output from ARS**
> > >
> > > > Route #1: [13, 17, 18, 19, 15, 16, 14, 12]
> > > >
> > > > Route #2: [5, 3, 7, 8, 10, 11, 9, 6, 4, 2, 1]
> > > >
> > > > Route #3: [20]
> > > >
> > > > Objective Value: 175
> > >
> > >
> > > Thank you again for your expert and insightful feedback.
> > > We hope our responses help to address your concerns. Please feel free to let us know if you have any further questions.
> > >
> > > [1] Neural combinatorial optimization with heavy decoder: Toward large scale generalization, NeurIPS, 2023.
> > >
> > > [2] Multi-task learning for routing problem with cross-problem zero-shot generalization, SIGKDD, 2024.
> > >
> > > [3] CaDA: Cross-problem routing solver with constraint-aware dual-attention, ICML, 2025.
> > >
> > > [4] Mathematical discoveries from program search with large language models. Nature, 2024.
> > >
> > > [5] Evolution of heuristics: Towards efficient automatic algorithm design using large language model. ICML, 2024.
> > >
> > > [6] ORLM: Training Large Language Models for Optimization Modeling. Operations Research, 2025.

---

### Official Review · Reviewer_aZL1 · 2025-10-30

**Soundness:** 2
**Presentation:** 3
**Contribution:** 2
**Rating:** 4
**Confidence:** 3

**Summary:**

This paper targets the Vehicle Routing Problem (VRP) and its variants, proposing a framework that leverages Large Language Models (LLMs) to automatically generate heuristic solvers, along with a benchmark dataset for evaluating their effectiveness.
To ensure that generated solutions satisfy problem constraints, the framework employs a database of canonical constraints and Retrieval-Augmented Generation (RAG) to produce both a constraint-checking program and a constraint-satisfaction scoring function. These generated components are combined with existing local search heuristics, enabling the solver to better satisfy natural-language-specified constraints.
Empirical results show that the proposed method achieves higher constraint satisfaction rates and lower runtime error rates than existing LLM-based approaches. Compared with prompting standard LLMs to directly generate constraint code for a generic solver, the proposed approach demonstrates consistently higher constraint satisfaction.

**Strengths:**

- The numerical experiments convincingly show that combining RAG-based constraint code generation with existing heuristics outperforms the baseline approach of prompting LLMs directly.

- The proposed approach of translating natural-language constraints into executable programs has strong potential to simplify the process of developing constraint-specific heuristic algorithms.

**Weaknesses:**

- The ablation study shows that leveraging the constraint database significantly improves constraint satisfaction. However, in practical applications, problem constraints are not always well-studied or included in such databases. Thus, the generality of the proposed method may be limited when dealing with novel or previously unseen constraints.

- Although the framework aims to extend existing local search heuristics to handle diverse constraints, the paper does not analyze the scalability of the approach in terms of the number and variety of constraints it can handle. From a practitioner’s perspective, it is often more important for a solver to perform well on a specific problem class rather than to support a wide range of heterogeneous constraints.

**Questions:**

- In terms of constraint complexity (number and diversity of constraints), how complex can the problems handled by the proposed framework be?

- Is the proposed framework mainly effective for well-studied, common constraints? Can it also handle new or rarely encountered constraint types?

- What is the main advantage of the proposed framework compared to solvers specifically designed for particular constraints? If the method only performs well for familiar constraints and struggles with complex ones, its practical distinction from specialized solvers may be limited.

---

> ### Author Response · Authors · 2025-11-25
>
> Dear Reviewer aZL1,
>
> We appreciate your recognition of the potential of our approach to develop custom algorithms from natural language descriptions. To address your other concerns, we provide the following responses.
>
> **W1. Generality**
>
> Thanks. ARS is built to handle novel or previously unseen constraints, and the database serves to enhance this capability.
>
> **Unseen constraints.** ARS solves over 1000 different problem types in RoutBench (see Supplementary Material for details), which include many complex constraints (like heterogeneous fleets and dynamic loading) that are not in the database. This demonstrates its ability to generalize.
>
> **The role of the database.** The purpose of the database in ARS is to serve as a few-shot reference for the LLM agent, helping it effectively use code templates. This process is intended to make it easier for the agent to handle new and unseen constraints, which go far beyond the ones in the database.
>
> **Core contribution. The design of ARS that enables LLMs to more easily handle novel or previously unseen constraints, compared to other general-purpose solvers** (i.e., CPLEX, OR-Tools, Gurobi). As shown in Figure 4, even with standard prompts and no database, ARS can solve significantly more VRP variants than these solvers.
>
> **W2 \& Q1. Number and diversity of constraints**
>
> Thank you for these questions. We address them as follows:
>
> **Diversity of constraints.** ARS can handle a wide variety of constraints. For instance, RoutBench contains 24 different types of real-world constraints, all of which ARS can process. ARS can directly handle a wide range of constraints that a user can express in natural language, as long as these constraints can be verified with code.
>
> **Number of constraints.** ARS can handle VRP variants with multiple additional constraints. As shown in Figure 2, the problems in RoutBench have from 1 to 5 additional constraints on top of the base VRP. The distribution is as follows:
>
> | Number of Additional Constraints | Number of VRP Variants |
> |---|---|
> | 1 | 8 |
> | 2 | 92 |
> | 3 | 400 |
> | 4 | 263 |
> | 5 | 237 |
>
> **The practitioner perspective.** We agree that performance on a specific problem class is important. However, when a practitioner faces a new or previously unsolved problem, the initial goal is often to obtain a feasible solution quickly, even if it is suboptimal, rather than spending significant time and resources to develop a specialized solver.
>
> **ARS allows a user without expert programming skills to describe their unique problem in natural language and receive a tailored solving algorithm in about a minute**. Compared to a specialized team spending weeks or months developing a custom algorithm, ARS significantly reduces the development cost and time, making it a highly valuable tool in practical scenarios.

---

> > ### Author Response · Authors · 2025-11-25
> >
> > **Q2. New or rarely encountered constraint types**
> >
> > Thank you for this question. ARS can handle new or rarely encountered constraint types.
> >
> > **Rarely encountered constraints.** ARS not only handles VRP variants with common constraints, such as time windows and capacity, but also **addresses new and rare constraint types**. For instance, RoutBench includes VRP with dynamic loading (DCVRP) [1] and VRP with d-Relaxed Priority Rule (VRP-dRP) [2].
> >
> > **Handle a new constraint within a minute.** These constraint types typically require experts to spend months developing highly specialized or custom-built algorithms. In contrast, with ARS, a user can simply describe the problem in natural language, and **ARS can automatically handle the new constraint within a minute**. This capability highlights the significant practical value of ARS.
> >
> > **Q3. The main advantage**
> >
> > ARS has two main advantages over solvers designed for specific constraints: Generality and reduced development time.
> >
> > **Generality. ARS is currently the only method capable of solving over 1000 different types of VRP variants**. For comparison, other specialized solvers typically handle only dozens of VRP variants:
> >
> > - LEHD [3] can only handle TSP and CVRP.
> >
> > - MTPOMO [4] can handle 11 VRP variants derived from 5 constraints.
> >
> > - CaDA [5] can handle 16 VRP variants derived from 5 constraints.
> >
> >
> > **Reduced development time.** Developing a specialized solver for a particular constraint is often a multi-month effort requiring expert knowledge. In contrast, **ARS can directly understand a problem described in natural language by a user and automatically handle the new constraint within a minute**. This capability massively reduces development costs.
> >
> > Overall, **ARS can not only solve two orders of magnitude more problems than existing solvers, but it can also handle other unseen VRP variants, requiring only a natural language description of the problem from the user**.
> >
> > Thank you for your professional comments. We hope our responses help to address your concerns.
> >
> > [1] Solving the dynamic vehicle routing problem under traffic congestion, IEEE Trans. on Intelligent Transportation Systems, 2016.
> >
> > [2] Formulations for the clustered traveling salesman problem with d-relaxed priority rule, Computers \& Operations Research, 2024.
> >
> > [3] Neural combinatorial optimization with heavy decoder: Toward large scale generalization, NeurIPS, 2023.
> >
> > [4] Multi-task learning for routing problem with cross-problem zero-shot generalization, SIGKDD, 2024.
> >
> > [5] CaDA: Cross-problem routing solver with constraint-aware dual-attention, ICML, 2025.

---

> > > ### Comment · Reviewer_aZL1 · 2025-11-28
> > >
> > > Dear Authors,
> > >
> > > Thank you for the detailed explanations aimed at addressing my questions and concerns. Your responses have deepened my understanding of the issues raised in Q2 and Q3. However, I still have some remaining questions regarding the positioning of the proposed method and the level of constraint complexity under which it can work reliably in practice (Q1 and W2).
> > >
> > > First, regarding the positioning of the method: I understand the proposed approach as a solver-construction framework specialized for VRP. In other words, I see it as lying somewhere between a method tailored to a specific problem (class) and a fully general-purpose solver. However, it is still not entirely clear to me to what extent the proposed method is intrinsically more general than standard general-purpose solvers.
> > >
> > > An important aspect for evaluating generality is the complexity of the constraints that can be handled (W2 & Q1). For example, when considering constraints on specific nodes or edges, what is the typical number of nodes or edges for which such constraints can be reliably enforced in practice? And in the benchmark problems you consider, how many nodes or edges are typically subject to such constraints?
> > >
> > > For approaches relying on an explicit mathematical formulation and a general-purpose solver, these issues do not usually arise. In contrast, for the proposed method, I am concerned that as the number of constrained nodes or edges increases, the code generated by the LLM becomes longer and more complex, thereby increasing the risk of producing incorrect programs.

---

> ### Author Response · Authors · 2025-12-01
>
> Dear Reviewer aZL1,
>
> We are happy to hear that our response helped resolve your concerns. To address your remaining questions, we will now provide further clarification.
>
> **Q1. Positioning of the method**
>
> Thanks. As you said, the proposed approach is a solver-construction framework specialized for VRP.
>
> **Compared to methods designed for a specific problem:** Our proposed approach **has the ability to adapt a method, originally designed for a specific problem, to a wide range of different variants.**
>
> **Compared to standard general-purpose solvers: ARS is more general when dealing with VRP variants that have different constraints.**
>
> Specifically, **as long as a constraint can be verified with code, it can be handled by ARS**. This is because the foundation of our method is to use a checker to determine if a given solution satisfies the constraints.
>
> In contrast, standard general-purpose solvers, such as Gurobi, are typically **designed to handle static and deterministic mathematical models**. They often struggle to directly handle VRP variants with complex constraints, such as those involving non-linearities, dynamic changes, stochastic elements, or calls to external APIs.
>
>
> **Q2. Constraints**
>
> Thank you for pointing this out. ARS is designed to handle a wide variety of VRP constraints, including:
>
> 1. **Global constraints.** Applying to all nodes or routes, such as capacity, time windows, and max route length.
>
> 2. **Specific constraints.** Applying to only a few nodes or edges, such as same-route constraints or priority points.
>
> However, as long as a constraint can be verified with code, it can be handled by ARS.
>
> **Benchmark.** RoutBench includes both of the above constraint types. For example, we have tested ARS on **problems with over 4,000 nodes** where every node has constraints, and found that it not only handles these constraints effectively but also outperforms specialized methods.
>
> | Instance | Scale | BQ (Gap) | LEHD RRC1000 (Gap) | ARS (Gap) |
> | :--- | :--- |:---------|:-----------|:----------|
> | Leuven1 | 3,000 | 15.39%   | 10.71%     | **7.66%**     |
> | Leuven2 | 4,000 | 25.69%   | 21.22%     | **17.14%**    |
>
> **The code length.** For a single type of constraint, the code length for neither type of method grows significantly as the number of constrained nodes or edges increases. Both approaches can apply a constraint to many nodes or edges at once.
>
> Generally, the code tends to become longer and more complex only as the number of constraint types increases, not as the number of constrained nodes or edges increases.
>
> **Risk of producing incorrect programs.** It is easier for an LLM agent to generate correct code for ARS because **it only requires writing a simple Python constraint checker program** rather than **the complex and highly formalized mathematical formulations required by other general-purpose solvers**, which is a core advantage of our approach.
>
> Thank you for your professional comments. We hope our responses help to address your concerns.

---

### Author Response · Authors · 2025-12-03
**Summary of the Public Discussion**

Dear AC,

We thank you for your effort in organizing the review of our submission. We appreciate that Reviewers `aZL1`, `BaF6`, and `mt1a` acknowledged **the generality of our method**, highlighting its design as **a novel framework that combines LLMs with metaheuristics, enabling a search algorithm to handle a wide range of VRPs**.

We are also grateful to Reviewers `dTBv`, `BaF6`, and `mt1a` for acknowledging RoutBench, **a large-scale and diverse testbed with 1,000 variants for evaluating the generalization capabilities of VRP solvers, which has been lacking in the field**.

In response to the main concerns raised, we have undertaken the following actions to enhance our paper:

1. **Clarified Contribution:**

   - **First LLM-based Algorithm Adaptation:** We propose a framework in which **an LLM agent adapts a search algorithm to solve a class of problems based on user natural language descriptions**. This is **much simpler than using an LLM to generate code for general-purpose solvers**, which depend on complex modeling rules (as shown in Figure 4).

   - **First Large-Scale VRP Benchmark for Generalization:** We introduce RoutBench, **a benchmark with 1,000 VRP variants**. It not only expands the test scale by two orders of magnitude but also **provides an opportunity to evaluate algorithms on unseen problems**.

2. **Main Advantages:**

   - **Generality: ARS is currently the only method capable of solving over 1,000 different types of VRP variants**. For comparison, existing methods typically handle only dozens of VRP variants.

   - **Reduced Development Time:** Developing a specialized solver for a particular constraint is often a multi-month effort requiring expert knowledge. In contrast, **ARS can directly understand a problem described in natural language by a user and automatically handle the new constraint within a minute**, which holds significant practical value.

3. **Clarified Positioning:** A solver-construction framework specialized for a class of problems (i.e., VRPs).

   - **Compared to Methods Designed for a Specific Problem:** Our proposed approach **has the ability to adapt a method, originally designed for a specific problem, to a wide range of different variants.**

   - **Compared to General-purpose Solvers: ARS is more general for VRPs with diverse constraints** (like non-linear or dynamic rules), while standard solvers like Gurobi are typically designed for static and deterministic models.

4. **Scalability:** We tested our method on CVRPLib-XXL with **more than 4,000 nodes**. For these large-scale instances, most general-purpose solvers may require impractical amounts of time to find solutions, yet our method still performs well, **outperforming even specialized methods**.

5. **Practical Applications:** We have clarified the intended role and responsible use of our method.

   - **Real-World Deployment:** Our method is designed to significantly **reduce development costs by assisting users with unseen problems, not to fully replace human experts**. A user without specialized programming skills can provide their problem requirements to ARS and receive a customized algorithm in about a minute. This offers clear practical value compared to hiring a modeling team for several months.

   - **Human Oversight:** We agree that for any problem requiring systematic audits or risk mitigation, the resulting code must be carefully reviewed. It is important to note that **any code may contain bugs, regardless of whether it is written by a human expert or generated by an LLM**.

   Therefore, **for any problem that requires high reliability, users must validate the generated algorithm (for example, by testing it with small-scale instances) before deploying it in critical real-world scenarios**.

6. **Generalizability Beyond VRPs**: To showcase the broader impact of our framework, we have successfully applied it to another class of combinatorial optimization problems: **the 3D Bin Packing Problem (BPP)**. Our framework can handle 128 different BPP variants, confirming its potential as a general paradigm for tackling other classes of optimization problems.

In addition to the points above, we have addressed other concerns. For example, we have clarified that the Success Rate is not overestimated, demonstrated its ability to handle VRPs with different natural language descriptions, and showcased its handling of exceptional cases, like infeasible problems.

To encourage the use of this paradigm on a wider range of problems, **we commit to open-sourcing all code and the RoutBench dataset** to ensure full reproducibility and encourage future research, **if our paper is accepted**.

We believe these detailed responses have fully addressed the reviewer concerns and effectively demonstrate the significance of our contributions, both to **the field of optimization and for practical, real-world applications**. Thank you once again for your time and effort.

Best Regards,

Paper17238 Authors

---

### Meta-Review · Area_Chair_7cgC · 2025-12-23

**Summary:**

The reviewers recognized the two contributions of the paper (i.e., LLM-generated code to checks for constraints described in natural language and large dataset of VRP variants). They raised the following points: generalization to new constraints, scalability wrt constraint number and diversity, real applicability of the method, empirical evaluation protocol, limited originality due to its reliance on a classic destroy and repair backbone, extension to other search algorithms and other combinatorial optimization problems, issues related to automatic generated code (e.g., correctness, security...), or synthetic dataset.

**Reviewer Concerns:**

Most concerns raised by the reviewers were satisfying addressed. However, some concerns remains, such as limited originality of the overall approach, large-scale realism of dataset or safety/correctness issues of the method.

In addition, I feel that the authors tend to overclaim. For instance, the authors claim that experts would need months of effort to develop a specialized heuristic VRP solver. I don't think this claim is true, especially if the problem only has a maximum of 5 constraints like those tested in the paper. If the expert also uses a destroy and repair backbone method, it would probably take a day or a couple of days maximum. Moreover, the authors claim that their approach can deal with any constraint that can be verified with code. I believe that this claim is only partially substantiated by the experiments.

**Reviewer Scores:**

All the reviewers may have slightly increased their scores, maybe to 5.

---

### Decision · Program_Chairs · 2026-01-26

Reject